# Motor biases reflect a misalignment between visual and proprioceptive reference frames

Tianhe Wang[1,2]*, J Ryan Morehead[3], Amber Jiang[1,2], Richard B Ivry[1,2†], Jonathan S Tsay[4*†]

[1]Department of Psychology, University of California, Berkeley, Berkeley, United States; [2]Department of Neuroscience, University of California, Berkeley, Berkeley, United States; [3]School of Psychology, University of Leeds, Leeds, United Kingdom; [4]Department of Psychology, Carnegie Mellon University, Pittsburgh, United States

*For correspondence:
shion07070017@gmail.com (TW);
jtsay2@andrew.cmu.edu (JST)

†These authors contributed equally to this work

## eLife Assessment

This **important** study uses an original method to address the longstanding question of why reaching movements are often biased. The combination of a wide range of experimental conditions and computational modeling is a strength. **Convincing** evidence is presented in support of the main claim that most of the biases in 2-D movement planning originate in misalignment between visuo-proprioceptive reference frames.

**Abstract** Goal-directed movements can fail due to errors in our perceptual and motor systems. While these errors may arise from random noise within these sources, they also reflect systematic motor biases that vary with the location of the target. The origin of these systematic biases remains controversial. Drawing on data from an extensive array of reaching tasks conducted over the past 30 years, we evaluated the merits of various computational models regarding the origin of motor biases. Contrary to previous theories, we show that motor biases produced by human participants do not arise from systematic errors associated with the sensed hand position during motor planning or from the biomechanical constraints imposed during motor execution. Rather, motor biases are primarily caused by a misalignment between eye-centric and body-centric representations of position. This model can account for motor biases across a wide range of contexts, encompassing movements with the right versus left hand, finger versus hand movements, visible and occluded starting positions, as well as before and after sensorimotor adaptation.

## Introduction

Accurate movements are crucial for everyday activities, affecting whether a glass is filled or spilled, or whether a dart hits or misses the target. Some movement errors arise from sensorimotor noise, including visual noise regarding the location of targets and effectors (*Burge et al., 2008*; *Tassinari et al., 2006*; *Osborne et al., 2005*), planning noise introduced when issuing a motor command, and neuromuscular noise when executing a movement (*Dideriksen et al., 2012*; *de C Hamilton et al., 2004*; *Dhawale et al., 2017*).

In addition, some of these errors arise from systematic biases that vary across the workspace (*Vindras et al., 1998*; *Gordon et al., 1994*; *Ghilardi et al., 1995*; *Vindras et al., 2005*). The origin of these biases remains controversial (*Figure 1a*): Whereas some studies postulate that motor biases stem from systematic distortions in perception (*Vindras et al., 1998*; *Vindras et al., 2005*; *Holden*

**Figure 1.** Different causes of motor biases. (**a**) Motor biases may originate from biases in perceiving the initial hand position (proprioceptive bias), perceiving the location of the visual target (target bias), transforming positional information from visual to proprioceptive space (transformation bias), and/or biomechanical constraints during motor execution. Previous models attribute motor biases to errors originating from the distinct contributions of visual (**b**) and proprioceptive biases (**c**). (**d**) Our model attributes motor biases to a transformation error between visual and proprioceptive coordinate systems. (**e**) A visuo-proprioceptive map showing the matching error between proprioceptive and visual space (*Wang et al., 2020*). Participants matched the position their hand (tip of the arrow) from a random starting location to the position of a visual target (end of the arrow). The blue dot depicts an example of a visual target in the workspace, and the red arrow indicates the corresponding matched hand position. Participants were asked to maximize spatial accuracy rather than focus on speed. (**f–h**) Simulated motor bias functions predicted by four models. Top: illustration of how each model yields a biased movement, with the example shown for a movement to the 135° target in panels **g** and **h** and for the 100° target in panel **f** (as there is no target bias at 135°). Gray bars in panels (**f–h**) indicate predicted bias for all targets and/or start position based on previous measurement of visual bias (**f**) (*Huttenlocher et al., 2004*), and proprioceptive/transformation bias (**g–h**) (*Wang et al., 2020*). Bottom: simulated motor bias functions differ qualitatively in terms of the number of peaks and troughs. Note that the middle panel depicts two variants of a proprioception model.

The online version of this article includes the following figure supplement(s) for figure 1:

**Figure supplement 1.** Previous studies have considered two variants of the Proprioceptive Bias model.

---

*et al., 2015*; *Yousif et al., 2024*; *Huttenlocher et al., 2004*), others posit that biases originate from inaccurate motor planning and/or biomechanical constraints associated with motor execution (*Goble et al., 2007*; *Alexander, 1997*; *Nishii and Taniai, 2009*). In the following section, we provide an overview of current models of systematic motor biases as well as outline a novel hypothesis, setting the stage for a reanalysis of published data and presentation of new experimental results.

Starting at the input side, motor biases may arise from systematic distortions in the representation of the perceived target position (*Figure 1b*). A prominent finding in the visual cognition literature is that the remembered location of a visual stimulus is biased toward diagonal axes (*Yousif et al., 2024*; *Huttenlocher et al., 2004*; *Kosovicheva and Whitney, 2017*). That is, the reported visual location of a stimulus is shifted towards the centroid of each quadrant. This bias does not depend on the

method of response, as this phenomenon can be observed when participants point to a cued location or press a key to indicate the remembered location of a briefly presented visual target (*Yousif et al., 2024*; *Huttenlocher et al., 2004*). While this literature has emphasized that this form of bias arises from processing within visual working memory, it is an open question whether it contributes to goal-directed reaching when the visual target remains visible.

Another potential cause of motor biases stems from proprioception (*Figure 1c*). Systematic distortions in the perceived position of the hand (*Rincon-Gonzalez et al., 2011*; *van Beers et al., 1998*; *Wang et al., 2020*) and/or joint position (*Sober and Sabes, 2005*; *Sober and Sabes, 2003*) can influence motor planning. For example, if the perceived starting position of the hand is leftwards of its true location, a reaching movement to a forward visual target would exhibit a rightward bias and a reaching movement to a rightward visual target would overshoot (*Vindras et al., 1998*). A proprioceptive perceptual bias at the starting position has been reported to be the major source of bias in many previous studies (*Rincon-Gonzalez et al., 2011*; *van Beers et al., 1998*; *Wang et al., 2020*; *Sober and Sabes, 2005*; *Sober and Sabes, 2003*).

Whereas the preceding models have considered how distortions of visual or proprioceptive space might, on their own, lead to reaching biases, reaching biases could also originate from a misalignment in the mapping between perceptual and motor reference frames. Based on classic theories of motor planning (*Buneo et al., 2002*), the start position and the target position are initially encoded in an eye-centric visual coordinate frame, and then transformed to representations in a body-centric proprioceptive coordinate frame within which the movement is planned (*Figure 1d*). Motor biases could arise from systematic distortions that occur during this visuo-proprioceptive transformation process (*Soechting and Flanders, 1989*; *Tillery et al., 1991*; *Flanders and Soechting, 1995*). Indeed, when participants are required to match the position of their unseen hand with a visual target, systematic transformation biases are observed across the workspace (*Figure 1d*; also see 'Methods'; *Jones et al., 2010*; *Cressman and Henriques, 2010*). In the current study, we develop a novel computational model to capture how these transformation biases should result in systematic motor biases during reaching.

On the output side, it has been posited that reaching biases could arise from biomechanical factors that impact movement execution (*Gordon et al., 1994*). Specifically, movements may be biased toward trajectories that minimize inertial resistance and/or energetic costs (*Alexander, 1997*; *Nishii and Taniai, 2009*; *Balasubramanian et al., 2009*; *Summerside et al., 2024*). For example, minimizing energy expenditure would result in biases toward trajectories that minimize resistive forces or changes in joint angles (*Goble et al., 2007*; *Soechting et al., 1995*). Moreover, inaccuracies in the internal model of limb dynamics could produce systematic execution biases. For example, underestimating the weight of the limb would result in reaches that overshoot the target (*Goble et al., 2007*; *Gordon et al., 1995*).

To determine the origin of motor biases, we formalized four computational models to capture the potential sources described above. As detailed in the 'Results' section, the models predict distinct motor bias patterns in a center-out reaching task (*Figure 1f–h*). While prior research has focused on the impact of individual sources (e.g., vision or proprioception) on the pattern of motor errors, these studies often entail a limited set of contexts (e.g., reaching behavior only when the start position is visible or only with the right hand). However, looking across studies, the task context can result in dramatically different motor bias patterns; indeed, when plotted in polar coordinates across the workspace, the bias functions range from having single peak (*Vindras et al., 1998*; *Vindras et al., 2005*; *Sober and Sabes, 2005*) to quadruple peaks (*Yousif et al., 2024*; *Huttenlocher et al., 2004*; *Kosovicheva and Whitney, 2017*). This diversity underscores the absence of a comprehensive explanation for motor bias phenomena across different experimental designs and setups. Additionally, a notable limitation of earlier work is the reliance on small participant cohorts (n<10) and a restricted number of targets (typically 8). The sensitivity of such experiments is limited in terms of their capacity to discriminate between models.

To better evaluate sources of motor bias during reaching, we report a series of experiments involving a range of contexts, designed to test predictions of the different models. We compared movements performed with the right or left hand, finger versus hand movements, under conditions in which the start position was either visible or not visible, and before and after implicit sensorimotor adaptation. To increase the power of model comparisons, we measured the motor bias function at a higher resolution (24 targets) and in a bigger sample size (n>50 per experiment).

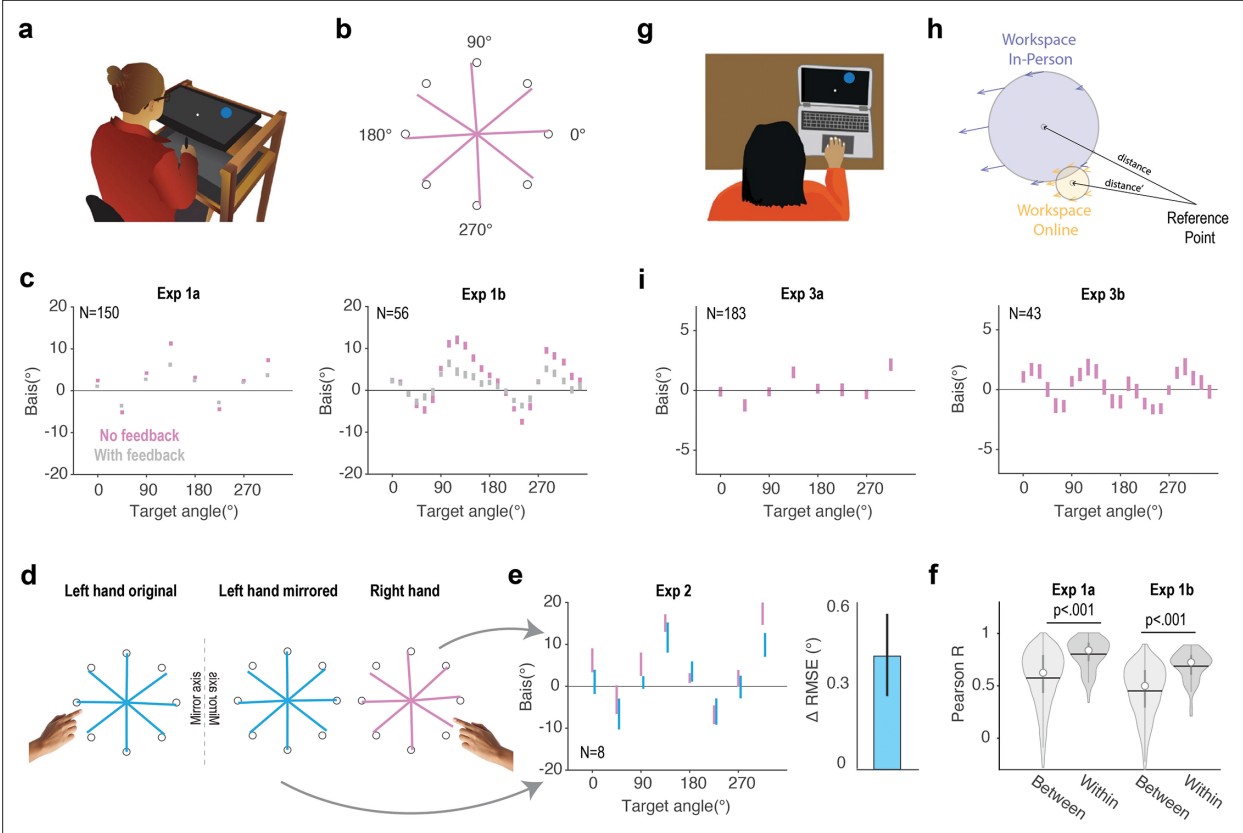

**Figure 2.** Motor biases across different experimental contexts. (**a**) Lab-based experimental apparatus for Exps 1–2. (**b**) Vectors linking the start position to the average endpoint position when reach amplitude equaled the target radius (pink lines; Exp 1a). (**c**) Motor biases as a function of target location. The dots indicate the mean angular error across participants during the no-feedback block (pink) and veridical feedback block (gray). The pattern of motor bias was similar in Exp 1a (8-targets; left panel) and Exp 1b (24-targets; right panel), characterized by two peaks and two troughs. Error bars denote standard error. (**d**) Motor biases generated during left hand reaches (left), left-hand results when the data are mirror reversed across the vertical meridian (middle), and right-hand reaches (right). (**e**) Left: mirror reversal of biases observed during left hand reaches is similar to biases observed with right hand reaches. Right: difference in RMSE when the right-hand map is compared to the original left-hand map relative to when the right-hand map is compared to the mirror-reversed left-hand map. Positive values indicate better data alignment when the left-hand data are mirror-reversed. (**f**) Correlation of the motor bias function between the no-feedback and feedback blocks is higher in the within-participant condition compared to the between-participant condition. Gray bars indicate the 25% and 75% quartiles. White dots indicate the median and horizontal lines indicate the mean. (**g**) Experimental setup for Exp 3. Participants were asked to make center-out reaching movements using a trackpad or mouse. These movements predominantly involve finger and wrist movements. (**h**) The workspace is presumed to be closer to the reference point (e.g., left shoulder) for finger/wrist movements (Exp 3) compared to that of arm movements (Exp 1–2). The transformation maps for the in-person and online spaces were simulated from the best-fit models in Exp 1 and Exp 2, respectively. (**i**) The pattern of motor biases in finger/wrist movements for 8-targets (left) and 24-targets (right).

## Results

### Motor biases across the workspace

To examine the pattern of motor biases during goal-directed movements, participants performed a center-out reaching task with their right hand (*Figure 2a*). We ran two versions of the study in Experiment 1. In Exp 1a, we used an eight-target version similar to that used in most previous studies (*Vindras et al., 1998*; *Gordon et al., 1994*; *Ghilardi et al., 1995*; *Vindras et al., 2005*; *Sober and Sabes, 2005*). To obtain better resolution of the motor bias pattern, we also conducted a 24-target version in Exp 1b. Within each experiment, participants first performed the task without visual feedback to establish their baseline bias and then a block with veridical continuous feedback to examine how feedback influences their biases. Motor biases were calculated as the angular difference between the target and hand when the movement amplitude reached the target distance (*Figure 2b*), with a positive error indicating a counterclockwise bias and a negative error indicating a clockwise bias.

Across the workspace, the pattern of motor biases exhibited a two-peaked function (*Figure 2c*) characterized by two peaks and two troughs. From the eight-target experiment, larger biases were

observed for the diagonal targets (45°, 135°, 225°, 335°) compared to the cardinal targets (0°, 90°, 180°, 270°) (*Slijper et al., 2009*; *van der Kooij et al., 2013*). In terms of direction, reaches to diagonal targets were biased toward the horizontal axis, and reaches for cardinal targets were biased in the counterclockwise direction. This pattern is similar to what has been observed in previous studies for right-handed movements (*Gordon et al., 1994*; *Ghilardi et al., 1995*). With the higher resolution in the 24-target experiment, we see that the peaks of the motor bias function are not strictly aligned with the diagonal targets but are shifted toward the horizontal axis. Moreover, the upward shift of the motor bias function relative to the horizontal line suggests that clockwise biases are more prevalent compared to counterclockwise biases across the workspace.

## Motor biases primarily emerge from a misalignment between visual and hand reference frames

We developed a series of models to capture how systematic distortions at different sensorimotor processing stages would cause systematic motor biases. Here we consider processing associated with the perceived position of the target, the perceived position of the arm/hand, and planning processes required to transform a target defined in visual space to a movement defined in arm/proprioceptive space. Biases could also arise from biomechanical constraints. Given that biomechanical biases are not easily simulated, we will evaluate this hypothesis experimentally (see below).

We implemented four single-source models to simulate the predicted pattern of motor bias for a center-out reaching task (*Figure 1f–h*; see 'Methods'). Since the task permits free viewing without enforced fixation, we assume that participants shift their gaze to the visual target; as such, an eye-centric bias is unlikely. Nonetheless, prior studies have shown a general spatial distortion that biases perceived target locations toward the diagonal axes (*Huttenlocher et al., 2004*; *Kosovicheva and Whitney, 2017*). Interestingly, this bias appears to be domain-general, emerging not only for visual targets but also for proprioceptive ones (*Yousif et al., 2024*). We incorporated this diagonal-axis spatial distortion into a Target Bias model. This model predicts a four-peaked motor bias pattern (*Figure 1f*).

For the Proprioceptive Bias model, we considered two variants building on the core idea that the perceived starting position of the hand is distorted. The first variant is a Vector-Based model in which the motor plan is a vector pointing from the perceived hand position to the target (*Vindras et al., 1998*; *Vindras et al., 2005*). The second variant is a Joint-Based model in which the movement is encoded as changes in the shoulder and elbow joint angles to move the limb from a start position to a desired target location (*Sober and Sabes, 2005*; *Sober and Sabes, 2003*; see *Figure 1*). Importantly, both models predict a motor bias function with a single peak (*Figure 1g*). Taken together, models that focus on systematic distortions of perceptual information do not qualitatively capture the observed two-peaked motor bias function (*Figure 2c*).

The fourth model, the Transformation Bias model, is based on the idea that the start and target positions are initially encoded in visual space and transformed into proprioceptive space for motor planning (*Buneo et al., 2002*). Motor biases may thus arise from a transformation error between these coordinate systems. Studies in which participants match a visual stimulus to their unseen hand or vice versa provide one way to estimate this error (*Rincon-Gonzalez et al., 2011*; *van Beers et al., 1998*; *Wang et al., 2020*; *Jones et al., 2010*). Two key features stand out in these data *Figure 1e*: First, the direction of the visuo-proprioceptive mismatch is similar across the workspace: For right-handers using their dominant limb, the hand is positioned leftward and downward from each target. Second, the magnitude of the mismatch increases with distance from the body. Using these two empirical constraints, we simulated a visual-proprioceptive error map (*Figure 1h*) by applying a leftward and downward error vector whose magnitude scaled with the distance from each location to a reference point. This model predicts a two-peaked motor bias function (*Figure 1h* Bottom), a shape strikingly similar to that observed in Exps 1a and 1b. Importantly, the model predictions are insensitive to the parameter values over a reasonable range. Thus, the number of peaks predicted by each model is a core distinguishing feature. We provide a theoretical analysis of how each model generates different bias functions in Appendix 1.

Note that the Proprioceptive Bias model and the Transformation Bias model tap into the same visuo-proprioceptive error map. The key difference between the two models arises in how this error influences motor planning. For the Proprioceptive Bias model, planning is assumed to occur in visual

space. As such, the perceived position of the hand (based on proprioception) is transformed into visual space. This will introduce a bias in the representation of the start position. In contrast, the Transformation Bias model assumes that visually-based representations of the start and target positions need to be transformed into proprioceptive space for motor planning. As such, both positions are biased in the transformation process. In addition to differing in terms of their representation of the target, the error introduced at the start position is in opposite directions due to the direction of the transformation (see *Figure 1g and h*).

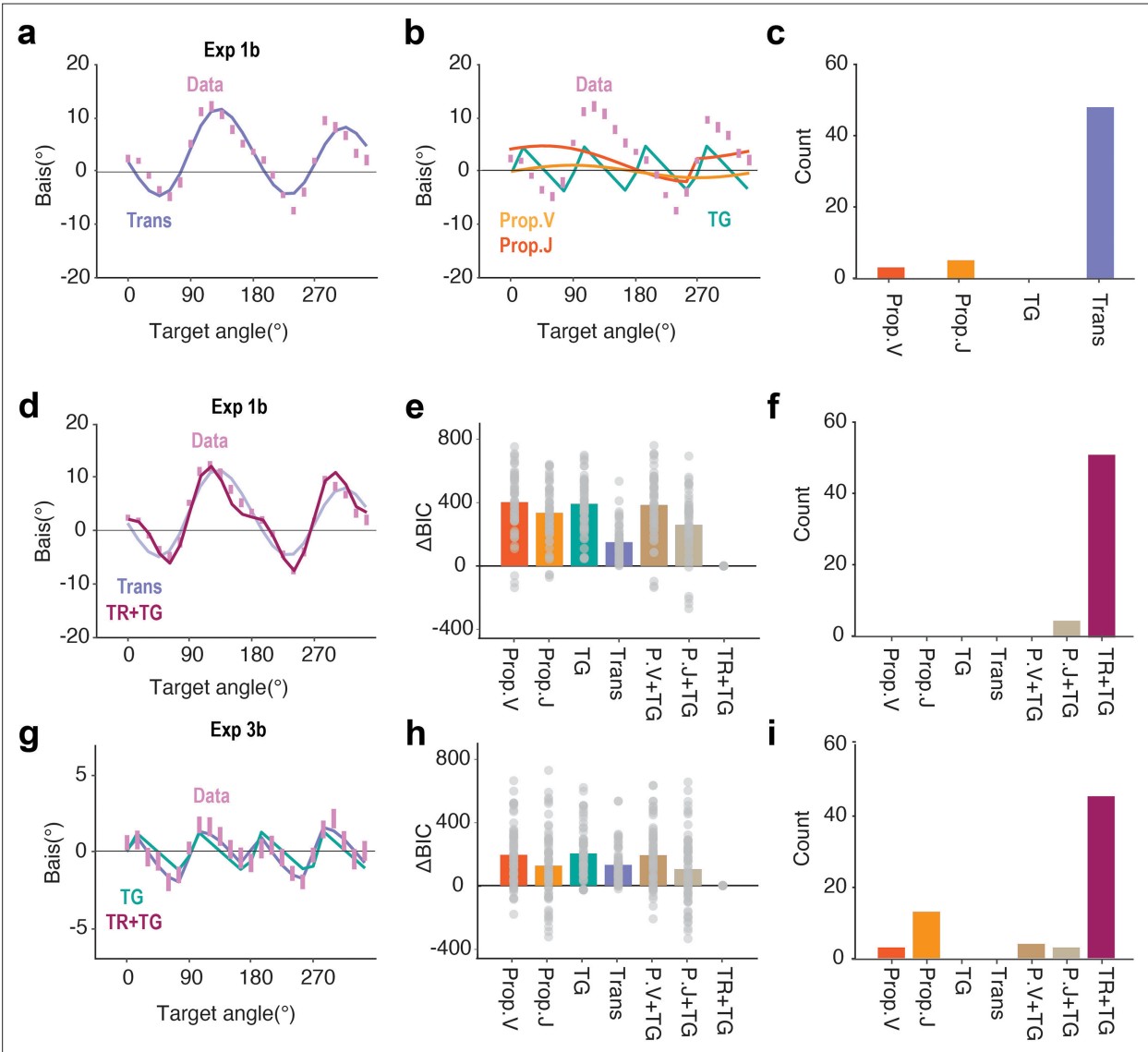

**Figure 3.** The pattern of motor biases is best explained by assuming systematic distortions in the perceived location of the target and the transformation between visual and proprioceptive coordinate frames. (**a**) For single-source models, the pattern of motor biases in the no feedback block of Exp 1b (pink dots) is best fit by the Transformation Bias model. (**b**) The three input-based models cannot explain the two-peak motor bias function. (**c**) Considering only the four single-source models, the data overwhelmingly favored the Transformation Bias model (48 out of 56 participants). (**d**) A mixture model involving transformation and target biases (TR +TG) provides the best fit to the motor bias function in Exp 1b (top). (**e**) Model comparison using BIC in Exp 1b. ΔBIC values are provided by subtracting the BIC from the best performing model (i.e., the TR +TG model). A smaller ΔBIC signifies better model performance. (**f**) For the mixture models, the data for almost all of the individuals were best explained by the TR + TG model (50 out of 56). (**g–i**) Same as panels (**d–f**), but for Experiment 3b.

The online version of this article includes the following figure supplement(s) for figure 3:

**Figure supplement 1.** Model recovery.

**Figure supplement 2.** Data and fits for four individuals.

To quantitatively compare the models, we fit each model with the data in Exp 1b at both the group and individual level. The Transformation Bias model provided a good fit of the two-peaked motor bias function ($R^2$=0.84, *Figure 3a*, see *Supplementary file 1* for parameters). *Figure 2h* shows the recovered visual-proprioceptive bias map based on the parameters of the Transformation Bias model when fit to the group-level reaching data in Exp 1b. The simulated results are very similar to the map measured empirically in a previous study (*Wang et al., 2020*; *Figure 1e*). In contrast, the Target Bias and Proprioceptive Bias models provide poor fits (all $R^2$ <0.18, *Figure 3b*). In terms of individual fits, the Transformation Bias model provided the best fit for most of the participants (48/56, *Figure 3c*). Thus, the model fitting results suggest that motor biases observed in reaching primarily originate from a transformation between visual and proprioceptive space.

A second way to evaluate the models is to compare the motor bias functions for the left and right hands. The Proprioceptive and Transformation Bias models predict that the bias function will be mirror-reversed for the two hands, whereas the Target Bias model predicts that the functions will be superimposed. We compared the functions for right and left hand reaches in Exp 2 using the eight-target layout. We found that the dissimilarity (quantified by the root mean square error, RMSE) between the pattern of motor biases across two hands significantly decreased when the left-hand data are mirror-reversed compared to when the bias patterns are compared without mirror reversal (t(78) = 2.7, p=0.008, *Figure 2d and e*). These results are consistent with the Transformation Bias model and provide further confirmation that the Target Bias model fails to provide a comprehensive account of reaching biases across both hands.

While the overall pattern of biases in the visuo-proprioceptive map is similar across individuals, there are subtle individual differences (see *Figure 3—figure supplement 2* for examples; *Rincon-Gonzalez et al., 2011*; *Wang et al., 2020*). As such, we would anticipate that the motor bias function would also exhibit stable individual differences due to idiosyncrasies in the visuo-proprioceptive map. To examine this, we correlated the bias functions obtained from blocks in which we either provided no visual feedback or veridical endpoint feedback. The magnitude of the biases was attenuated when endpoint feedback was provided, likely because the feedback reduced the visuo-proprioceptive mismatch. Nonetheless, the overall pattern of motor bias was largely preserved, with the within-participant correlations (Exp 1a: $r_{norm}$ = 0.999, Exp 1b: $r_{norm}$= 0.974) significantly higher than the averaged between-participant correlation in both Exp 1a (0.569) and Exp 1b (0.455) (*Figure 2f*).

## Target bias also contributes to the motor bias

In the preceding section, we considered each model in isolation, testing the idea that motor biases arise from a single source. However, the bias might originate from multiple sources. For example, there could be a distortion in both vision and proprioception, or a visuo-proprioceptive transformation that operates on distorted inputs. To address this, we evaluated hybrid models by combining the Target Bias model with the Proprioceptive or Transformation Bias models. Although theoretically plausible, we did not consider a hybrid of the Proprioceptive and Transformation Bias models since they directly conflict in terms of whether the start position is perceived visually or proprioceptively.

The hybrid model that combines the Transformation and Target Bias models (TR+TG model) provided an excellent fit of the motor bias pattern in Exp 1b ($R^2$=0.973, *Figure 3d*). Based on a comparison of BIC values, this model not only outperforms the other hybrid models but also significantly improved the fit compared to the Transformation Bias model alone (*Figure 3e and f*). These results are especially interesting in that the assumed target bias toward the diagonal axes has only been shown in studies in which perception was tested after the target had been extinguished. The current results suggest that this bias is also operative when the target remains visible, suggesting that the target bias may reflect a general distortion in how space is represented, rather than a distortion that arises as information is processed in visual working memory.

To further evaluate the TR+TG model, we examined its performance in explaining the motor bias function obtained in an on-line study (Exp 3) in which participants performed the center-out task by moving a finger across a trackpad. One major difference between the in-person and online setups is that the workspace is much smaller and closer to the body when participants use a trackpad (*Figure 2g*). As such, the magnitude of the motor biases generated by transformation errors should be smaller with the online setup (Exp 3) compared to the in-person setup (Exp 1–2; *Figure 2h*).

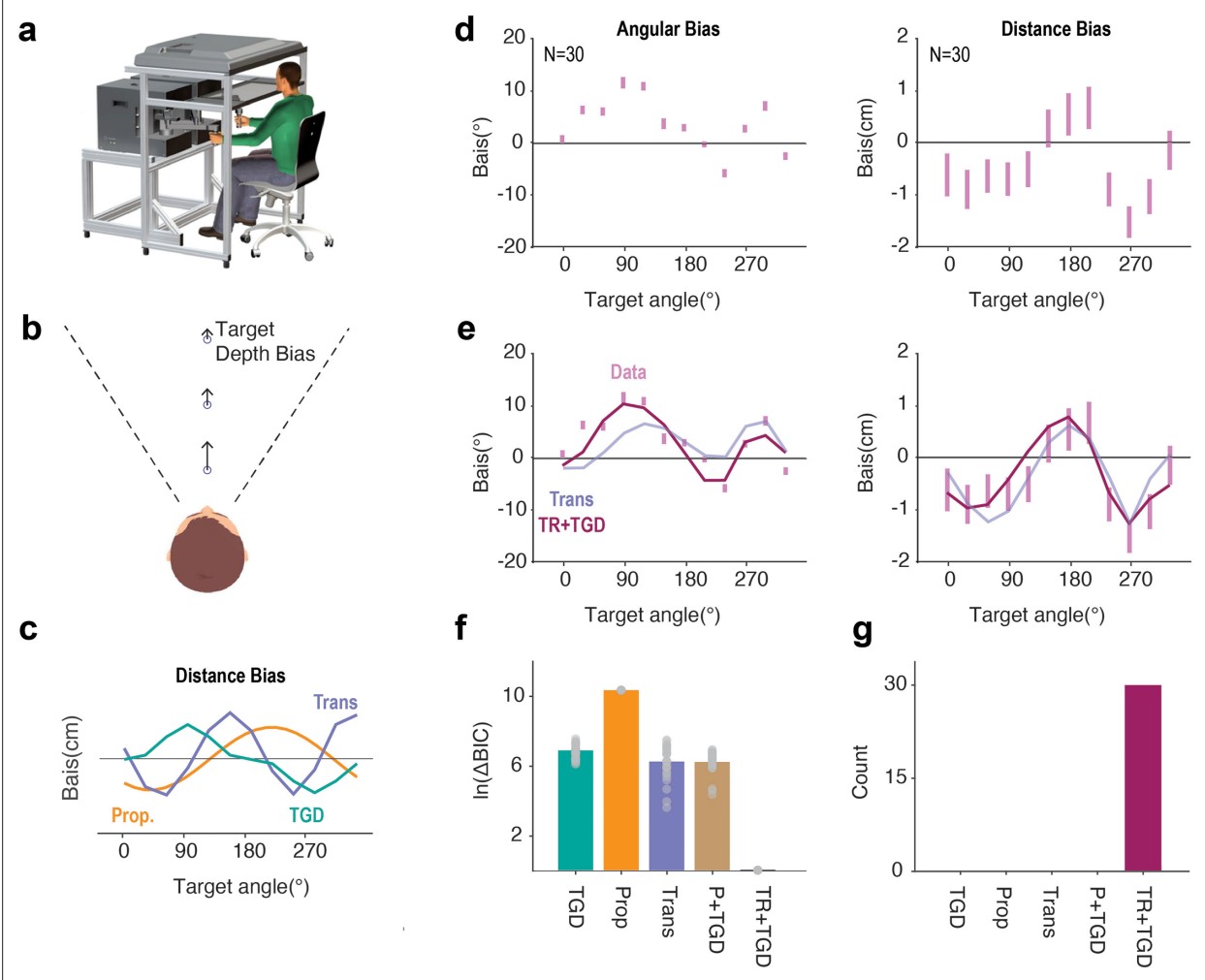

**Figure 4.** Motor biases in both angular and distance dimensions originate from a misalignment between visuo-proprioceptive reference frames. (**a**) KINARM apparatus for Exp 4. (**b**) Assuming that participants viewed the display from a fixed angle, we would expect a perceptual bias in depth perception (*Volcic et al., 2013*; *Hibbard and Bradshaw, 2003*). (**c**) Model simulations for motor biases in movement extent. The Transformation Bias model predicts a two-peaked function for distance bias, while the Proprioceptive Bias and the Target Depth Bias (TGD) models predict a one-peaked function. (**d**) Participants exhibited a two-peaked bias function for reach angle and extent. (**e**) The hybrid Transformation Bias+Target Depth Bias (TR+TGD) provides a good fit to the data for both dimensions. (**f, g**) Model comparisons. The TR+TGD model outperformed alternative models in terms of averaged ΔBIC (**f**) and model frequency (**g**).

Consistent with the prediction of the TR+TG model, we found markedly smaller motor biases with the online setup *Figure 2i* (Exp 3) compared to the in-person setup (Exp 1) (*Figure 2c*). While the overall shape of the motor bias functions was similar across experiments, we observed two small peaks between 20° and 200° in Exp 3 that were not apparent in Exp 1. When we fit this function to single source models, the Target Bias model outperformed the Transformation Bias model. This suggests that when the movements are close to the body, target biases make a relatively stronger contribution to the motor biases compared to transformation biases. Nonetheless, the TR+TG model again provides the best fit to the motor bias function ($R^2$=0.857, *Figure 3h*, see *Supplementary file 1* for best-fit parameter values), significantly outperforming all other alternatives (*Figure 3i*).

## Motor biases in movement distance

To this point, we have focused on how reaching biases are manifest in angular error. However, a complete model should also account for biases measured in terms of radial error (i.e., distance). To examine the source of both angular and radial errors, we conducted a center-out reaching task and

instructed the participants to rapidly move their hand, attempting to terminate the movement directly on the target rather than slice through the target (*Figure 4a*).

We again simulated each model to generate predictions, now for both angular and radial errors. In its simplest form, the Target Bias model does not make specific predictions about distance biases given that prior work has focused exclusively on angular errors and lacked data on distance-related target biases. However, in our KINARM setup, participants likely viewed targets from a fixed angle (*Figure 4a and b*). This would introduce a perceptual bias in depth (*Volcic et al., 2013*; *Hibbard and Bradshaw, 2003*). To account for this, we simulated a variant, the Target Depth (TGD) Bias model. This model predicts single-peaked bias functions for angular and radial errors. The Proprioceptive Bias model makes a similar prediction. (Note: We used the vector-based version, as the joint-based model does not generate predictions for movement extent.) In contrast, the Transformation Bias model predicts a two-peaked bias pattern in both the angular and distance dimensions (*Figure 4c*).

The reaching biases exhibited two peaks when the error is plotted for both angle and extent (*Figure 4d*). We jointly fit the 2D bias data using the three single-source models and all corresponding mixture models (*Figure 4e*). Similar to our prior results, the Transformation Bias model was the best single-source model and the combination of the Target Bias and the Transformation Bias model outperformed all other models, best explained all participants (*Figure 4g*). Together, the data from all four experiments converge on the conclusion that motor biases are primarily driven by a misalignment between visuo-proprioceptive reference frames.

## Transformation bias model accounts for qualitative changes in the motor bias function

The Transformation Bias model assumes that, for normal reaching, both the start and the target positions are encoded in visual space before being transformed into proprioceptive space for motor planning. However, if the start position is not visible, then the sensed start position would be directly encoded in proprioceptive space (i.e., where the hand is positioned), bypassing the need for a transformation between coordinate frames. As such, biases arising from the transformation process would only arise when the input is limited to the perceived position of the visual target. When we simulated the scenario in which the start position is not visible, the Transformation Bias model predicts a single-peaked function (*Figure 5a*, right), a qualitative change from the two-peaked function predicted when both the start position and target position are visible (*Figure 5a*, left).

To test this idea, we re-examined data from previous studies in which the participant's hand was passively moved to a start position with no visual information given about the start location or hand position (*Vindras et al., 1998*; *Vindras et al., 2005*; *Sober and Sabes, 2005*). Strikingly, the motor bias function under this condition has only one peak (*Figure 5b*). Thus, the Transformation Bias model provides a novel account of the difference in motor biases observed when the start position is visible (Exp 1–3) compared to when it is not visible.

We note that the one-peaked motor bias function has previously been interpreted as evidence in support of a Proprioceptive Bias model (*Figure 1g*; *Vindras et al., 1998*; *Vindras et al., 2005*; *Sober and Sabes, 2005*). We performed a model comparison on the data from one of these studies (*Vindras et al., 2005*) and the TR+TG outperformed the Proprioceptive Bias model, as well as the Prop+TG models (ΔBIC=10.9). In addition, only the TR+TG model can account for the asymmetry between clockwise and counterclockwise biases. In summary, these results suggest that motor biases when reaching from an unseen start position arise when the target position is transformed from visual to proprioceptive coordinates rather than from a proprioceptive bias impacting the sensed start position. Moreover, the TR+TG model provides a parsimonious account of the bias functions, independent of the visibility of the start position.

Another way to evaluate the Transformation Bias model is to perturb the position of the visual start position relative to the real hand position. Under this manipulation, a single-peaked motor bias function is observed (*Figure 5—figure supplement 1*; *Sober and Sabes, 2005*; *Sober and Sabes, 2003*). Interestingly, the functions exhibit opposing phase shifts when the starting position is perturbed to the left versus to the right (*Figure 5—figure supplement 1*). This qualitative change in the motor bias function can again be successfully captured by the Transformation Bias model. Taken together, these data provide strong evidence favoring the notion that motor biases originate from a misalignment between visuo-proprioceptive reference frames.

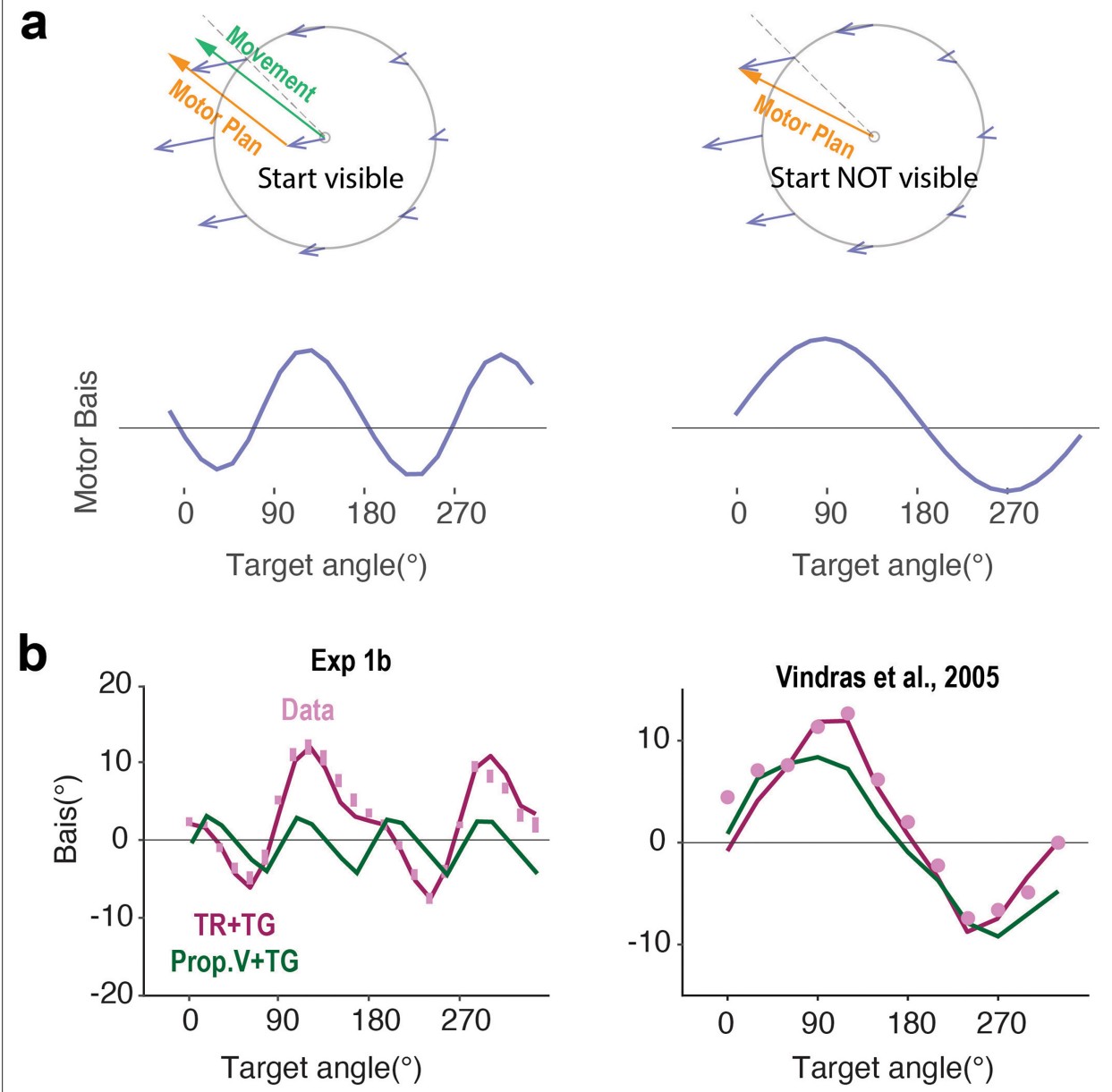

**Figure 5.** Motor bias pattern changes when the start position is not visible. (**a**) Schematic showing the planned movement under the Transformation Bias model when the start position is either visible (left) or not visible (right). In the latter case, only the target position is transformed from visual to proprioceptive coordinates with the start position directly encoded in proprioceptive space. The TR+TG model now predicts a single-peaked motor bias function (lower row). (**b**) Consistent with this prediction, a two-peaked function is predicted when the start position is visible (as in Exp 1), and a single-peaked function is predicted when start position is not displayed. Data (pink dots) are from *Vindras et al., 2005*.

The online version of this article includes the following figure supplement(s) for figure 5:

**Figure supplement 1.** The Transformation Bias model can explain the motor bias functions when the visual information is shifted.

## Biomechanical models fail to account for motor biases

An alternative account of motor biases is that they arise from biomechanical constraints associated with upper limb movement. For example, movement kinematics have been explained in terms of cost functions that minimize energy expenditure and/or minimize jerk (*Flash and Hogan, 1985*). These constraints might result in an increase in endpoint error (*Alexander, 1997*; *Nishii and Taniai, 2009*; *Balasubramanian et al., 2009*). To evaluate the effect of biomechanical constraints on reach accuracy, we used a state-of-the-art biomechanical model of the upper limb, the MotorNet (*Codol et al.,*

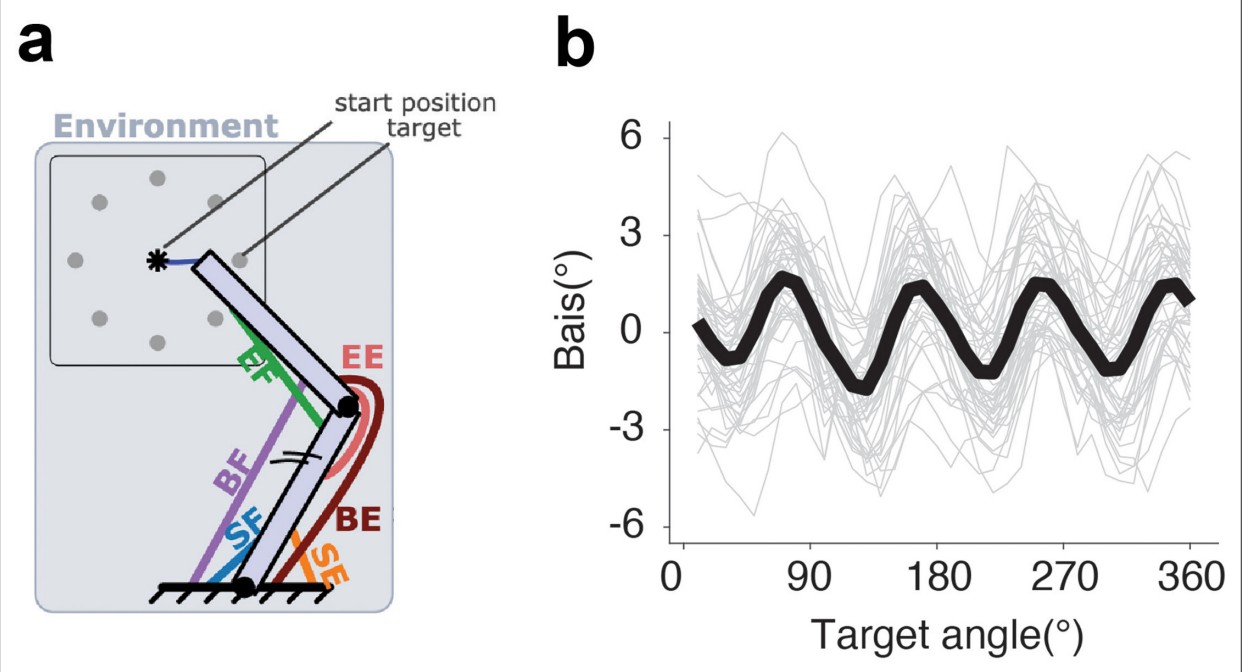

**Figure 6.** Biomechanical constraints are unlikely to be a primary source of motor biases. (**a**) Schematic of the two-skeleton, six-muscle effector used in MotorNet and how predictions concerning reaching biases were simulated (adapted from *Codol et al., 2024*). (**b**) The model predicts a four-peaked motor bias function for a center-out reaching task, at odds with the two-peaked functions observed in Exps 1–3. Gray lines denote single simulations. Black line denotes the group average across runs.

*2024*), to simulate motor biases in a center-out reaching task. This model is composed of two bones and six muscle actuators, with the control policy generated by a recurrent neural network (*Figure 6a*, see 'Methods'). While the model captures basic biomechanical constraints of the upper limb (*Codol et al., 2024*), it produces a four-peaked angular bias function (*Figure 6b*). Thus, the model fails to capture the empirically observed two-peaked function. This simulation suggests that biomechanical constraints are unlikely to be a primary source of motor biases.

As a second comparison of the TR+TG and Biomechanical Bias models, we examined how motor biases change after the sensorimotor map is recalibrated following a form of motor learning, implicit sensorimotor adaptation. Here we reanalyzed the data from previous experiments that had used a perturbation technique in which the visual feedback was always rotated by 15° from the target, independent of the hand position (*Figure 7a*, non-contingent clamped feedback; *Morehead et al., 2017*). Participants adapt to this perturbation, with subsequent reaches to the same target shifted in the opposite direction (*Figure 7b*), reaching an asymptote of around 20° and showing a robust aftereffect when the perturbation is removed. Participants are unaware of their change in hand angle in response to clamped feedback, reporting their perceived hand position to be close to the target (*Tsay et al., 2020*).

For the TR+TG model, the transformation between visual and proprioceptive space depends on the perceived positions of the start and target locations in a visual-based reference space, one that remains *unchanged* before and after adaptation. We assume that adaptation has changed a senso-rimotor map that is referenced after the transformation from visual to proprioceptive space. As such, the heading angle after adaptation for each target location is obtained by summing the motor biases for that target location and the extent of implicit adaptation. This would result in a vertical shift of the motor bias function (*Figure 7c*, top).

In contrast, the biomechanical model predicts that motor biases will be dependent on the actual movement direction rather than the target location (e.g., a bias toward a movement that is energeti-cally efficient). Since the mapping between a target location and its corresponding reach direction is rotated after adaptation, the motor bias pattern would also be rotated (*Figure 7c*, bottom). As such,

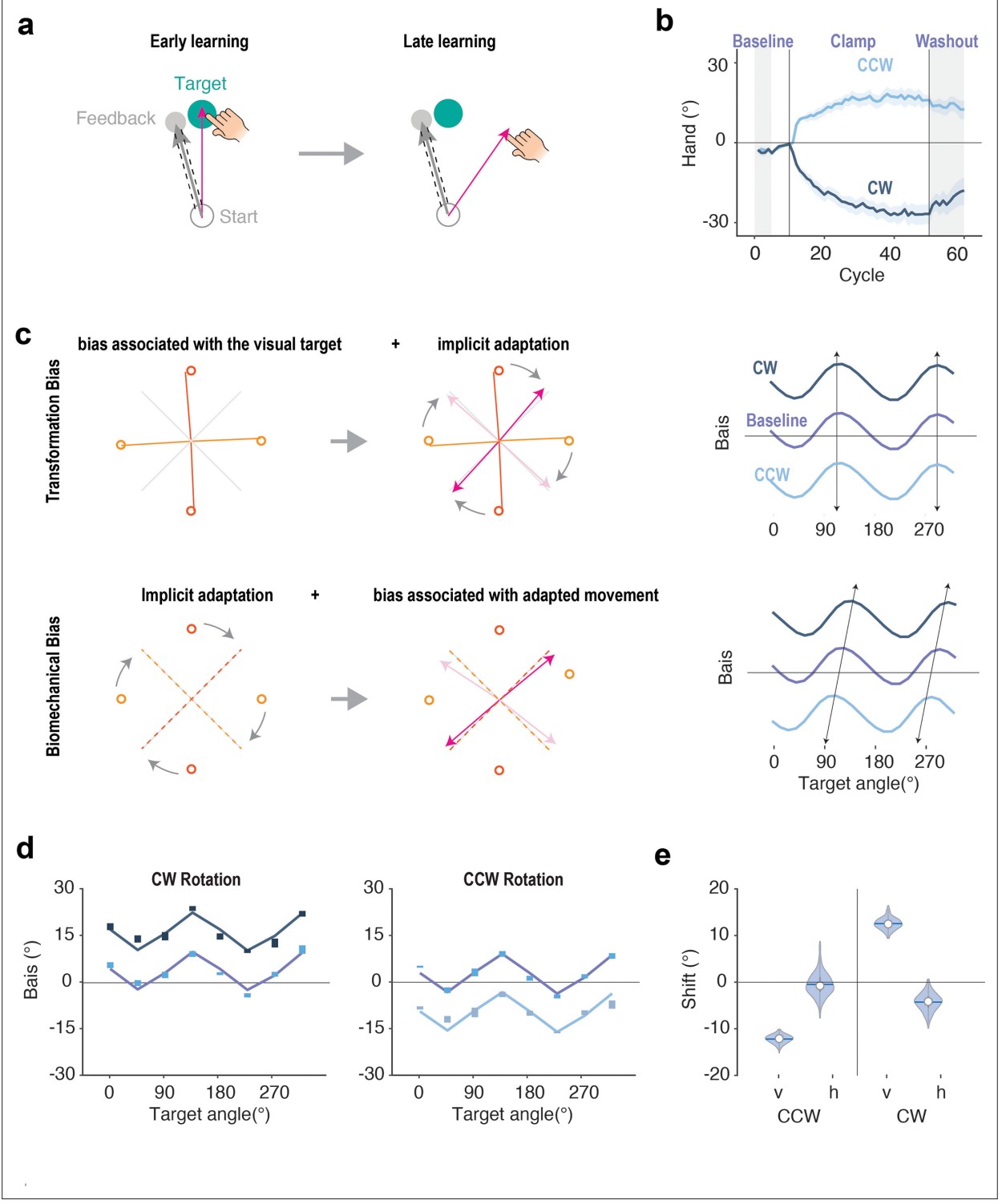

**Figure 7.** The pattern of motor bias is preserved after implicit sensorimotor adaptation, consistent with the Transformation+Target Bias model. (**a**) Illustration of the clamped perturbation. Feedback cursor is offset by a fixed angle from the target, independent of the participant's heading direction. (**b**) Time course of hand angle in response to clockwise or counterclockwise clamped feedback. Vertical lines demarcate the perturbation block which was preceded and followed by no-feedback baseline and washout phases, respectively (gray areas). Shaded area indicates standard error. (**c**) Predictions for the bias functions after adaptation for the TR+TG (top) and Biomechanical models (bottom). See text for details. The right column shows the predicted motor bias functions following adaptation in response to a clockwise (CW) or counterclockwise (CCW) clamp. (**d**) Motor bias functions before and after training in a CW (left) and a CCW (right) clamp. Data taken from *Morehead et al., 2017* and *Kim et al., 2018*; the height

*Figure 7 continued on next page*

*Figure 7 continued*

of the colored bars indicates the standard error for each data point. The best-fit lines for the TR+TG model are shown. (**e**) Parameter values to capture vertical and horizontal shifts in motor bias functions before and after training. The CW and CCW conditions both showed a significant vertical shift but no horizontal shift.

The online version of this article includes the following figure supplement(s) for figure 7:

**Figure supplement 1.** Illustration of the TR+TG model.

the biomechanical model predicts that the motor bias function will be shifted along both the horizontal and vertical axes.

To arbitrate between these models, we analyzed the data from two previous studies, looking at the bias function from no-feedback trials performed before (baseline) and after adaptation (washout) (*Morehead et al., 2017*; *Kim et al., 2018*). Consistent with the prediction of the TR+TG model, the motor bias function shifted vertically after adaptation (*Figure 7d*) but did not shift horizontally.

To quantitatively evaluate these results, we first fit the motor bias function during the baseline phase with the TR+TG model and fixed the parameters. We then examined the heading angles during the aftereffect phase by fitting two additional parameters, one that allowed the function to shift vertically (*v*) and the other to allow the function to shift horizontally (*h*). The TR +TG model predicts that only *v* will be different than zero; in contrast, the Biomechanical Bias model predicts that *h* and v will both be different than zero and should be of similar magnitude. The results clearly favored the TR+TG model (*Figure 7d and e*). The vertical shift in the bias functions was of a similar magnitude as the aftereffect, with the shift direction depending on the direction of the clamped feedback (v: CW: 12.5°; CCW: –12.2°, p<0.001). In contrast, the best-fitting value for *h* was not significantly different from zero in both conditions. These results are consistent with the hypothesis that visual representations are first transformed into proprioceptive space for motor planning, with the recalibrated sensorimotor map altering the trajectory selected to achieve the desired movement outcome.

## Discussion

While motor biases are ubiquitous in goal-directed reaching movements, the origin of these biases has been the subject of considerable debate. We addressed this issue by characterizing these biases across a range of experimental conditions and evaluated a set of computational models derived to capture different possible sources of bias. Contrary to previous theories, our results indicate that motor biases do not stem from a distortion in the sensed position of the hand (*Vindras et al., 1998*; *Vindras et al., 2005*; *Sober and Sabes, 2005*; *Sober and Sabes, 2003*) or from biomechanical constraints during movement execution (*Alexander, 1997*; *Nishii and Taniai, 2009*; *Balasubramanian et al., 2009*). Instead, motor biases appear to arise from systematic distortions in perceiving the location of the visual target and the transformation required to translate a perceived visual target into a movement described in proprioceptive coordinates (*Soechting and Flanders, 1989*; *Tillery et al., 1991*; *Flanders and Soechting, 1995*). Strikingly, our model successfully accounts for sensorimotor biases across a wide range of contexts, encompassing movements performed with either hand as well as with finger versus hand movements. Our model also accounts for the qualitative changes in the motor bias function that are observed when vision of the starting position of the hand is occluded, and when the sensorimotor map is perturbed following implicit adaptation.

While motor biases have been hypothesized to reflect a mismatch across perceptual and motor coordinate systems (*Tillery et al., 1991*; *Flanders and Soechting, 1995*), it is unclear what information is transformed and what reference frame is employed for motor planning. Interestingly, many previous studies posit that movement is planned in an eye-centric visual reference frame (*Batista et al., 1999*; *Blohm et al., 2009*; *Henriques et al., 1998*). While the target can be directly perceived in this reference space, the start position of the hand would need to be transformed from a proprioceptive reference frame to a visual one. Systematic error in this transformation would mean that the start position of the hand is inaccurately represented in visual space, resulting in motor biases (*Vindras et al., 1998*; *Tillery et al., 1991*). This idea underlies the Proprioceptive Bias models described in this paper.

In contrast to these models, our Transformation Bias model posits that movement is planned in a hand-centric proprioceptive reference frame. By this view, when both the target and start position are provided in visual coordinates, the sensorimotor system transforms these positions from visual space

to proprioceptive space. Systematic error in this transformation process will result in motor biases. When vision of the start position is available, the Transformation Bias model successfully accounts for the two-peaked motor bias function (Exp 1). Even more compelling, the Transformation Bias model accounts for how the pattern of motor biases change when the visibility of the start position is manipulated. When the start position is occluded, the transformation from visual to proprioceptive space is only relevant for the target position since the start position of the hand is already represented in proprioceptive space. Here, the model predicts a motor bias function with a single peak, a function that has been observed in previous studies (*Vindras et al., 1998*; *Vindras et al., 2005*).

We note that there is a third scenario, one in which both the start position and target position are provided in proprioceptive space. We predict that under this condition, motor biases originating from the visuo-proprioceptive transformation would completely disappear. Indeed, when the hand is passively moved first to the target location and then to the start position, subsequent reaches to the target do not show the signature of bias from a visuo-proprioceptive transformation (*Yousif et al., 2024*). Instead, the reaches exhibited a bias toward the diagonal axes, consistent with the predicted pattern if the sole source of bias is visual.

Our TR+TG model extends beyond simple shooting movements and simultaneously accounts for movement biases for both angular and amplitude (extent) dimensions. Prior work (*Gordon et al., 1994*) attributed the bias in amplitude to movement inertia: rotating the elbow (i.e., moving along the axis orthogonal to the upper arm) entails lower effective inertia than moving along the axis parallel to the upper arm. Given the arm posture at the start position, the upper limb points toward 135°/315°, with the orthogonal direction corresponding to 45°/225°. The speed profiles in both our Exp 4 and *Gordon et al., 1994* are consistent with this hypothesis. However, this hypothesis does not specify a mechanism by which direction-dependent speed translates into directional extent biases. In brief, they effectively assume that endpoint extent bias mirrors speed bias. Yet the profile of the bias in movement extent does not follow the speed bias. For example, in our data, the speed function peaks around 45°, which corresponds to a valley in the extent bias function. As such, the transformation bias and visual-target bias likely play a larger role in determining the bias observed in movement endpoints.

Why would a sensorimotor system exhibit inherent biases during the transformation process? We propose that these biases arise from two interrelated factors. First, these systems are optimally tuned for distinct purposes: A body-centric system predominantly uses proprioceptive and vestibular inputs to determine the orientation and position of the body in space, while an eye-centric system relies on visual inputs to interpret the layout of objects in the external world, representations that should remain stable even as the agent moves about in this environment (*Proske and Gandevia, 2012*; *Héroux et al., 2022*). Second, these sensory systems consistently receive information with very different statistical distributions (*Tassinari et al., 2006*; *Zhang et al., 2015*), perhaps because of these distinct functions. For example, visual inputs tend to cluster around the principal axes (horizontal and vertical) (*van den Berg et al., 2012*; *Hahn and Wei, 2024*), whereas proprioceptive information during reaching is clustered around diagonal axes (*Mawase et al., 2018*). This is because these movements are often the least effortful and are the most frequently enacted directions of movement (*Shadmehr et al., 2016*). The differences in computational goals and input distributions might have led to natural divergences in how each system represents space (*van Beers et al., 2002*), and consequently, result in a misalignment between the reference frames.

The Transformation Bias model addresses how biases arise when the information is passed along from a visual to a proprioceptive reference frame. However, the results indicate that another source of bias originates from a distortion within the visual reference frame itself, manifesting as an attractive bias toward the diagonal axes. Thus, the best-fitting model posits two sources of bias, one related to the representation of the visual target and a second associated with the transformation process. This hybrid Transformation + Target Bias model outperformed all single-source and hybrid models, providing an excellent fit of the behavioral data across a wide variety of contexts.

What might be the source of the visual bias in the perceived location of the target? In the perception literature, a prominent theory has focused on the role of visual working memory account based on the observation that in delayed response tasks, participants exhibit a bias toward the diagonals when recalling the location of visual stimuli (*Huttenlocher et al., 2004*; *Sheehan and Serences, 2023*). Underscoring that the effect is not motoric, this bias is manifest regardless of whether the response

is made by an eye movement, pointing movement, or keypress (*Kosovicheva and Whitney, 2017*). However, this bias is unlikely to be dependent on a visual input as similar diagonal bias is observed when the target is specified proprioceptively via the passive displacement of an unseen hand (*Yousif et al., 2024*). Moreover, as shown in the present study, a diagonal bias is observed even when the target is continuously visible. Thus, we hypothesize that the bias to perceive the target toward the diagonals reflects a more general distortion in spatial representation rather than being a product of visual working memory.

Other forms of visual bias may influence movement. Depth perception biases could contribute to biases in movement extent (*Beurze et al., 2006*; *Van Pelt and Medendorp, 2008*). Visual biases toward the principal axes have been reported when participants are asked to report the direction of moving targets or the orientation of an object (*Wei and Stocker, 2015*; *Patten et al., 2017*). However, the predicted patterns of reach biases do not match the observed biases in the current experiments. We also considered a class of eye-centric models in which participants overestimate the radial distance to a target while maintaining central fixation (*Beurze et al., 2006*; *Van Pelt and Medendorp, 2008*). At odds with this hypothesis, participants undershot rightward targets when we measured the radial bias in Exp 4. The absence of these other distortions of visual space may be accounted for by the fact that we allowed free viewing during the task.

Our data suggest that biomechanical factors do not significantly impact motor biases. We provided several lines of evidence suggesting the biomechanical factors have minimal influence on the pattern of motor biases. For example, it is hard to envision a biomechanical model that would account for the qualitative change in the bias function when the start position was visible (two-peak function) to when it was hidden (one-peak function). More directly, simulations with a state-of-the-art biomechanical model produced motor bias patterns that did not resemble the empirical results.

We also evaluated biomechanical contributions to motor biases by examining the bias pattern observed before and after implicit sensorimotor adaptation. We assume that adaptation mainly modifies a sensorimotor map (*Tsay et al., 2022*) but has a relatively smaller influence on a visuo-proprioceptive map (*Cressman and Henriques, 2010*; *Tsay et al., 2020*; *Cressman and Henriques, 2009*). That is, adaptation may change the mapping between a target represented in the proprio-ceptive space and the motor commands required to reach that location. Given that a biomechanical model assumes that motor biases are associated with the direction of a movement, this model would predict that the pattern of motor biases would be distorted by implicit motor adaptation. At odds with this prediction, the pattern of motor biases remained unchanged after adaptation, a result consistent with the Transformation Bias model.

Nonetheless, the current study does not rule out the possibility that biomechanical factors may influence motor biases in other contexts. Biomechanical constraints may have had limited influence in our experiments due to the relatively modest movement amplitudes used and minimal interaction torques involved. Moreover, while we have focused on biases that manifest at the movement endpoint, biomechanical constraints might introduce biases that are manifest in the movement trajectories (*Alexander, 1997*; *Nishii and Taniai, 2009*). Future studies are needed to examine the influence of context on reaching biases.

## Methods

### Participants

For the lab-based study (Exps 1, 2, 4), 266 undergraduate students (age: 18–24) were recruited from University of California, Berkeley. For the online study (Exp 3), 183 young adult participants (age: 18–30) were recruited via Prolific, a website designed to recruit participants for online behavioral testing. All participants were right-handed as assessed by the Edinburgh handedness test (*Oldfield, 1971*) with normal or corrected-to-normal vision. Each participant was paid $20/h. All experimental protocols were approved by the Institutional Review Board at the University of California, Berkeley (approval number: 2016-02-8439). Informed consent was obtained from all participants.

## Procedure

### Experiments 1a, 1b, and 2

Experiments 1a, 1b, and 2 were conducted in the lab. Participants performed a center-out reaching task, holding a digitizing pen in the right or left hand to make horizontal movements on a digitizing tablet (49.3 cm × 32.7 cm, sampling rate = 100 Hz; Wacom, Vancouver, WA). The stimuli were displayed on a 120 Hz, 17-in. monitor (Planar Systems, Hillsboro, OR), which was mounted horizontally above the tablet (25 cm), to preclude vision of the limb. The experiment was controlled by custom software coded in MATLAB (The MathWorks, Natick, MA), using Psychtoolbox extensions, and run on a Dell OptiPlex 7040 computer (Dell, Round Rock, TX) with Windows 7 operating system (Microsoft Co., Redmond, WA).

Participants made reaches from the center of the workspace to targets positioned at a radial distance of 8 cm. The start position and target location were indicated by a white annulus (1.2 cm diameter) and a filled blue circle (1.6 cm), respectively. The vision of the hand was occluded by the monitor, and the lights were extinguished in the room to minimize peripheral vision of the arm. Feedback, when provided, was in the form of a 4 mm white cursor that appeared on the computer monitor, aligned with the position of the digitizing pen.

To start each trial, the participant moved the cursor to the start circle (5 mm diameter). After maintaining the cursor within the start circle for 500 ms, a target appeared at one of the target locations. The participant was instructed to make a rapid slicing movement through the target. We did not impose any reaction time guidelines, allowing the participant to set their own pace to initiate the movement. On no-feedback trials, the cursor was blanked when the hand left the start circle, and the target was extinguished once the radial distance of the movement reached the target distance (8 cm). On feedback trials, the cursor was visible throughout the movement until the movement amplitude reached 8 cm; at that point, its position was frozen for 1 s, providing feedback of the accuracy of the movement (angular position with respect to the target). After this interval, the target and cursor were extinguished.

At the end of both the no-feedback and feedback trials, a white ring appeared denoting the participant's radial distance from the start position. This ring was displayed to guide the participant back to the start position without providing angular information about hand position. Once the participant moved within 2 cm of the start position, the ring was extinguished, and a veridical cursor appeared to allow the participant to move their hand to the start position. If the amplitude of the hand movement did not reach the target (<8 cm radial distance) within 300ms, the message 'too slow' would be displayed for 500ms before the white ring appeared.

For Exps 1a and 2, there were eight target locations, evenly spaced in 45° increments around the workspace (primary axes and main diagonals). For Exp 1b, there were 24 target locations, evenly spaced in 15° increments. Each experiment consisted of a no-feedback block followed by a feedback block. There were five trials per target (40 trials total) for each block in the Exps 1a. There were four trials per target (96 trials total) in Exp 1b.

### Experiments 3a and 3b

Experiments 3a and 3b were conducted using our web-based experimental platform (*Tsay et al., 2021*). Participants made center-out movements by controlling a cursor with the trackpad on their personal computers. It was not possible to occlude vision of the hand. However, since the visual stimulus was presented on a vertical monitor and the hand movement was in the horizontal plane, we assume vision of the hand was limited to the periphery (based on observations that the eyes remain directed to the screen during the trial). The size and position of visual stimuli were scaled based on each participant's screen size (height = 239.6 ± 37.7 mm, width = 403.9 ± 69.5 mm). The experiment was controlled by custom software written with JavaScript and presented on Google Chrome. Data were collected and stored using Google Firebase.

The procedure was designed to mimic the lab-based experiments. On each trial, the participant made a center-out planar movement from the start position to a visual target. A white annulus (1% of screen height in diameter, 0.4 cm on average) indicated the start position, and a blue circle (1% of screen height in diameter) indicated the target location. The radial distance of the target from the start position was 40% of the screen height (5 cm on average). At the beginning of each trial, participants moved the cursor (0.6% of the screen height in diameter) to the start position, located at the

center of their screen. The cursor was only visible when its distance from the start position was within 20% of the screen height. After maintaining the cursor at the start position for 500 ms, the target appeared. The participant made a rapid slicing movement through the blue target. As in the online experiments, there were feedback and no-feedback trials. For feedback trials, the cursor was visible until it reached the target distance and then froze for 1 s at the target distance. On no-feedback trials, the cursor was extinguished after the hand exited the start position and the target disappeared once the radial distance of the movement reached the target distance. 500 ms after the end of the trial, the cursor became visible, repositioned at a random location within 10% of the screen height from the start position. The participant then moved the cursor to the start position to trigger the next trial.

There were eight target locations in Exp 3a and 24 target locations in Exp 3b. As with the lab-based experiments, each experiment included a no-feedback block followed by a feedback block. We obtained larger data sets in the online studies: For each block, there were 20 trials/target (160 total trials for Exp 3a and 480 total trials for Exp 3b).

## Experiment 4

To examine motor biases for both reach angle and extent, we performed a lab-based experiment with the KINARM system (BKIN Technologies). Participants performed a center-out reaching task, while holding onto the handle of a two-link robotic manipulandum. Vision of the arm and hand was occluded by a semi-silvered mirror that reflected the visual display from an LCD monitor mounted above the mirror (LG47LD452C, LG Electronics, 47 in., 1920 × 1080 pixel resolution). A black cloth was draped over the participant's shoulder and arm, and the lights were extinguished in the room to minimize peripheral vision of the arm. The participant was seated in a comfortable chair with their forehead resting against a soft leather patch at the height of the monitor. Kinematic data were recorded at a sampling rate of 1000 Hz, and with a spatial resolution of 0.1 mm. The experiment was controlled by a Dell OptiPlex 7040 computer (Dell, Round Rock, TX) running Dexterit-E software and coded in MATLAB Simulink.

On each trial, a red target (1 cm diameter) appeared at one of 12 locations, spaced 30° apart and positioned 10 cm from the start location (green circle; 1 cm diameter). After maintaining their hand within the home position for 500 ms, the target appeared. The participant was instructed to make a rapid, straight movement to the target. Unlike in Experiment 1, they were told to attempt to stop on the target, holding the position until the target disappeared. The instructions emphasized that they should not make corrective movements but rather, that they try to reach the target with the initial movement. The target disappeared 1 s after movement onset, defined as when the movement speed >0.01 cm/s. Hand position was recorded once the movement speed dropped below 1 cm/s. On 99.8% of trials, movement speed did not increase once this threshold was passed, indicating that the participants adhered to the instructions. On the remaining trials, we detected a secondary corrective movement (increase in speed>5 cm/s). On these trials, we used the position recorded when the movement speed initially dropped below 1 cm/s as the endpoint position. The pattern of results would be the same were we to exclude these trials. The robotic arm returned the hand to the central start position. No visual feedback was provided at any point during the experiment. Each block consisted of one reach to each of the 12 targets in randomized order, and the experiment was composed of 10 blocks.

## Reanalysis of prior data sets

*Vindras et al., 2005*. This study used a design in which the participant did not see the start position of the movement. This was achieved by not including start position information in the visual display and passively moving the participant's hand to a start position prior to each reach. Once positioned, a visual target would appear and the participant reached to that location. Across trials, there were two start positions, 12 target positions (spaced evenly by 30° around the workspace), and two target distances (6 and 12 cm). In modeling these data, we used the movement endpoint averaged across start positions and target distances.

*Morehead et al., 2017*; *Kim et al., 2018*. We reanalyzed the data from the 15° conditions of Exp 4 in *Morehead et al., 2017* and Exps 1 and 2 in *Kim et al., 2018*. These three experiments examined visuomotor adaptation using non-contingent clamped feedback. On perturbation trials, the feedback cursor was presented at the radial position of the hand but with a fixed 15° angular offset relative to the target. Participants were informed that the angular position was not contingent on their hand

position and instructed to move directly to the target, ignoring the feedback. This method results in robust implicit adaptation, with the heading direction of the movement gradually shifting away from the target in the opposite direction of the cursor. Participants are unaware of this change in behavior (*Tsay et al., 2020*). In each experiment, there were three blocks: A no-feedback baseline block (10 trials/target), a clamped feedback block (60 trials/target), and a no-feedback washout block (10 trials/ target).

## Data analyses

Motor bias refers to the angular difference between the position of the hand and target when the hand reaches the endpoint target distance. Angular errors were plotted as a function of the target position with 0° corresponding to the rightward target (3 o'clock location) and 90° corresponding to the forward target. Positive bias values indicate a counterclockwise error, and negative values indicate a clockwise error. Trials with errors larger than 3 standard deviations from each individual's mean error were removed from the analysis (<0.5%).

To assess the similarity of the motor bias functions across different conditions, we calculated the normalized correlation coefficient as $r_{norm} = \frac{r_{data}}{r_{max}}$. $r_{data}$ is the Pearson correlation coefficient between the two motor bias functions. $r_{max}$ is the correlation coefficient between the recorded motor bias function and the true (but unknown) underlying motor bias function from that condition. To calculate $r_{max}$, we used a method developed to measure the noise ceiling for EEG/fMRI data (*Schoppe et al., 2016*):

$$r_{max} = \sqrt{\frac{2}{1 + \sqrt{\frac{1}{r_{half}^2}}}}$$

where $r_{half}$ is determined by splitting the data set (based on participants) into random halves and calculating the correlation coefficient between the first half and the second half of the data. We bootstrapped $r_{half}$ by resampling the data 2000 times and used the average value. $r_{max}$ is calculated separately for a pair of conditions and the smaller one is applied as the normalizer for $r_{norm}$.

## Models

To examine the source of motor bias, we considered five single-source models and three multiple-source models.

### Target bias model

The Target Bias model postulates that movement biases arise because the perceived position of the visual target is systematically distorted (*Figure 1b*). Here we draw on the work of *Huttenlocher et al., 2004*. In their study, a visual target was picked from an invisible circle, presented for 1 s and then blanked. The participant then indicated the remembered position of the target by pointing to a position on a circular digitizing pad. The results showed a bias toward the four diagonal directions (45°, 135°, 225°, 315°), with the magnitude of this bias increasing linearly as a function of the distance from the diagonals. As such, the maximum bias was observed for targets close to four cardinal target locations (0°, 90°, 180°, 270°), and the sign of bias flipped at the four cardinal target locations.

We used the shape of this function to model bias associated with the perception of the location of the visual targets. To obtain a continuous function, we assumed a transition zone around the cardinal targets, each with a half-width represented by the parameter *a* (*Figure 1a*), and the peak motor bias is represented by the parameter *b*. As such, the angular bias (*y*) at a target located at *x*° can be formalized as

$$x' = x \bmod (90) \tag{1}$$

$$y = \frac{b * x'}{a}, if x' < a; \tag{2}$$

$$y = b - \frac{(x' - a) b}{45° - a}, if 90 - a > x' > a;$$

$$y = \frac{-b * \left(90 - x^{'}\right)}{a}, if\ x^{'} > 90 - a;$$

This model has two free parameters (*a* and *b*). If participants directly reach the perceived target location, their motor biases will directly reflect their visual biases.

## Vector-based proprioceptive bias model

(*Vindras et al., 1998*; *Vindras et al., 2005*) proposed a model in which movement biases result from a misperception in estimating the initial position of the hand (*Figure 1c*). Specifically, it has been shown that the perceived position of the hand when placed near the center of the workspace is biased toward the ipsilateral side and away from the body (*Rincon-Gonzalez et al., 2011*; *van Beers et al., 1998*; *Jones et al., 2010*). Assuming that the planned movement is formed by a vector pointing from the sensed hand position to the visual target position, this proprioceptive distortion will result in systematic motor biases around the workspace. For example, for the target at 90°, misperceiving the initial position of the right hand to the right of the start position will result in a movement that is biased in the counterclockwise (leftward) direction.

To simulate this Proprioceptive Bias model, we assumed the participants perceived the start position (0, 0) as a rightward bias away from the midline position, defining a proprioceptive error vector $(x_e, y_e)$. For a target i at $[x_i, y_i]$, the motor plan is a vector $[x_i - x_e, y_i - y_e]$. From this, we calculated the angular difference between the motor plan vector and the target position to generate the motor bias for each target. The two free parameters in this model are $[x_e, y_e]$.

## Joint-based proprioceptive bias model

Reaching movements may also be planned in joint coordinates rather than the hand (endpoint) position (*Sober and Sabes, 2005*; *Sober and Sabes, 2003*). Based on this hypothesis, motor biases could come about if there is a misperception of the initial elbow and shoulder joint angles. To implement a Joint-Based Proprioceptive Bias model, we represent the length of the forearm and upper arm as $l1$ and $l1$, respectively. We denote the initial angles of the shoulder and elbow joints as $\theta_o$ and $\varphi_o$, respectively, and their associated perceived error as $\theta_e$, and $\varphi_e$ (See *Figure 1—figure supplement 1*).

By setting the origin of the coordinate system for the right shoulder at $P_0$ (0, 0), the hand can be represented as

$$P\left(\theta, \varphi\right) = \left[l1cos_0 + l1cos_0, l1sin_0 + l2sin_0\right]. \tag{3}$$

For a fixed position in the workspace, there will be a unique solution pair for $\theta$ and $\varphi (\pi > \varphi > \theta > 0)$, should a solution exist. To calculate the required change in joint angle to reach a visual target, we assumed that the system plans a movement based on the perceived hand position:

$$P_p = P\left(\theta_0 + \theta_e, \varphi_0 + \varphi_e\right) \tag{4}$$

Then we solve the following equation to decide the proper $\sigma\theta_i$ and $\sigma\varphi_i$ that transfer the hand from the start position to a target *i* at $[x_i, y_i]$:

$$P_p - \left[x_i, y_i\right] = P\left(\theta_0 + \theta_e + \sigma\theta_i, \varphi_0 + \varphi_e + \sigma\varphi_i\right) - P_p \tag{5}$$

We calculated the real movement direction based on the real hand position:

$$\sigma h_i = P_p - P\left(\theta_0, \varphi_0\right) \tag{6}$$

We compare the directionof $h_i$ and the target direction to calculate the motor bias. For simplicity, we assume $l1 = l2 = 24$ cm (*Fryar et al., 2012*). The four free parameters in this model are $\theta_0$, $\varphi_o$, $\theta_e$, and $\varphi_e$.

## Transformation bias model

The Transformation Bias model proposes attributes motor biases to systematic errors that arise during the transformation from a visual to proprioceptive-based reference frame. To implement this model, we refer to an empirically derived visuo-proprioceptive error map from a data set that sampled most

of reachable space (**Figure 1d**, **Wang et al., 2020**). Specifically, in that study, participants were asked to move their unseen hand from a random start position to a visual target. Rather than require a discrete reaching movement, they were told to continuously adjust their hand position, focusing on accuracy in aligning the hand with the target. The direction of the error was relatively consistent across targets, with the final hand position shifted to the right and undershooting the target. The magnitude of these biases increased as the radial extent of the limb increased. This basic pattern has been observed across studies using different visuo-proprioceptive matching methods (**Rincon-Gonzalez et al., 2011**; **van Beers et al., 1998**; **Jones et al., 2010**; **Cressman and Henriques, 2010**; **Cressman and Henriques, 2009**; **Johnson et al., 2008**).

The matching errors provide an empirical measure of the transformation from a visual reference frame to a proprioceptive reference frame. To model these data, we defined a transformation error vector, $[x_e, y_e]$, whose direction is fixed across space. We then defined a 'reference position' with a coordinate of $[x_r, y_r]$. For upper-limb movements, this reference position is often considered to be positioned around the shoulder (**Haggard et al., 2000**). The transformation error vector at position $i$ is scaled by its Euclidean distance ($d$) to the referent position:

$$T_i = d_i[x_e, y_e] \tag{7}$$

$$d_i = \sqrt{\left(x_i - x_r\right)^2 + \left(y_i - y_r\right)^2} \tag{8}$$

Movements toward a target $i$ are planned via the vector connecting the start position to the target in proprioceptive space, denoted as

$$\sigma h_i = T_0 - \left(T_i + \left[x_i, y_i\right]\right), \tag{9}$$

where $T_0$ is the transformation vector at the start position, which is set as $[0,0]$. Motor bias is calculated as the angular difference between the motor plan and the target. The four free parameters in the Transformation Bias model are $x_e, y_e, x_r, y_r$.

## Visual depth bias model

Reaching movements were made with the KINARM system in Exp 4. With this system, the targets appear at an oblique angle (**Figure 4a**) and this may introduce perceptual biases in depth ($b_{y,i}$) (**Volcic et al., 2013**; **Hibbard and Bradshaw, 2003**).

$$b_{y,i} = ky_i + c \tag{10}$$

where $c$ is a consistent bias applied to all targets, and $k$ determines the specific bias for target $i$ based on its position on the y-axis ($y_i$) (**Figure 4b**). We assumed that the same rule for visual depth bias would apply to both the target and start position.

## Hybrid models

The four models described above each attribute motor biases to a single source. However, the bias might originate from multiple processes. To formalize this hypothesis, we considered three hybrid models, combining the Target Bias model with the two versions of the Proprioceptive Bias model and with the Transformation Bias model. We did not create a hybrid of the Proprioceptive and Transformation Bias models since they make different assumptions about the information used to derive the motor plan.

## Proprioceptive bias+target bias (P+TG) model

We also created two hybrid models, combining the Target Bias model with the Vector-Based and Joint-Based Proprioceptive Bias models. The Target Bias model is used to estimate systematic error in the perceived location of the target, and the Proprioceptive Bias models are used to estimate systematic error in the perceived position of the hand at the start position. For these models, we calculated the biases from two models separately and then added them together:

$$b_i\left(P + TG\right) = b_i\left(\text{Proprioceptive Bias}\right) + b_i\left(\text{Target Bias}\right), \tag{11}$$

where $b_i$ refers to the bias at target $i$.

## Transformation bias+target bias (TR+TG) model

To simulate the TR+TG model, the representation of the visual target is first determined based on the Target Bias model. The biased visual target is then transformed into proprioceptive coordinates based on the Transformation Bias model (*Figure 7—figure supplement 1*).

## Biomechanical model

To simulate motor biases that might arise from biomechanical constraints, we used MotorNet, a biomechanical model of the upper limb (*Codol et al., 2024*). For upper limb reaching, MotorNet includes two bones and six muscle actuators (ReluMuscle), and a control policy dictated by simple recurrent neural networks using PyTorch. We created a policy network architecture with the recurrent layer of the model simulated as a single layer of 32 gated recurrent units (GRUs). The output layer of the model is a fully connected linear layer with sigmoid nonlinearity. The network receives input concerning target position and the current state of effectors to generate a transformation matrix as a policy.

We trained the model to perform point-to-point reaching where the start and target positions were randomized across the whole workspace. We then examined the directional biases produced by the model in a center-out reaching task. Following the procedure recommended by the authors of MotorNet, we trained the model using mini-batch gradient descent with Adam optimizer, a learning rate of 1000, and 6000 batches of size 32. For each batch, the policy network was trained on a random reach to minimize L1 loss between the effector's fingertip trajectory and the goal trajectory. The model was evaluated with a center-out reaching task in which the targets were positioned at a radial distance of 10 cm from the center, mimicking the procedure in Exp 4. We used 36 target positions separated by 10°. Angular bias was measured as the angular difference between the simulated fingertip position at 80% of the trajectory distance and the intended target angle. Note that the model does not generate any prediction on movement extent in the current format. To estimate variability of the motor bias predictions, we trained and tested 50 independent instances of the model.

## Model comparison

To compare the models, we fit each model with the data from Experiments 1b and 3b in which reaches were made to 24 targets. We used the fminsearchbnd function in MATLAB to minimize the negative sum of loglikelihood (LL) across all trials for each participant. LL were computed assuming normally distributed noise around each participant's motor biases:

$$LL = normpdf(x, b, c) \tag{12}$$

where $x$ is the empirical reaching angle, $b$ is the predicted motor bias by the model, $c$ is motor noise, calculated as the standard deviation of $x$-$b$.

For model comparison, we calculated the BIC as follows:

$$BIC = -2LL + k * \ln(n) \tag{13}$$

where $k$ is the number of parameters of the models. Smaller BIC values correspond to better fits. We report the sum of ΔBIC by subtracting the BIC value of the TR+TG model from all other models.

For illustrative purposes, we fit each model at the group level, pooling data across all participants to predict the group-averaged bias function.

## Modeling motor bias after implicit sensorimotor adaptation

To examine how the motor bias function changes after visuomotor adaptation, we first used the TR+TG model to fit the motor bias function from a no-feedback baseline block tested prior to the introduction of the perturbation. We then used the best-fitted baseline model ($TrTgB$) to estimate the shift in the motor bias function from data obtained in a no-feedback aftereffect block following adaptation:

$$b(i) = TrTgB(i + v) + h \tag{14}$$

where $b(i)$ is the motor bias at target $i$ in the aftereffect; $v$ and $h$ indicate the vertical and horizontal shift, respectively. To estimate the distribution of $v$ and $h$, we bootstrapped the subjects with repetition for 200 times and fitted the $v$, $h$ based on the group average of each bootstrapped sample.

## Acknowledgements

We thank Zixuan Wang and Anisha Chandy for helpful discussions. We thank Anisha Chandy for data collection. RBI is funded by the NIH (grants NS116883 and DC017091). JST was supported by the NIH (F31NS120448).

## Additional information

### Competing interests

Richard B Ivry: is a co-founder with equity in Magnetic Tides, Inc. The other authors declare that no competing interests exist.

### Funding

| Funder | Grant reference number | Author |
|---|---|---|
| National Institute of Neurological Disorders and Stroke | DC017091 | Richard B Ivry |
| Foundation for Physical Therapy Research | F31NS120448 | Jonathan S Tsay |
| National Institute of Neurological Disorders and Stroke | NS116883 | Richard B Ivry |

The funders had no role in study design, data collection and interpretation, or the decision to submit the work for publication.

### Author contributions

Tianhe Wang, Conceptualization, Data curation, Formal analysis, Investigation, Visualization, Methodology, Writing – original draft, Project administration, Writing – review and editing; J Ryan Morehead, Conceptualization, Data curation, Formal analysis, Supervision; Amber Jiang, Data curation, Formal analysis, Visualization; Richard B Ivry, Conceptualization, Supervision, Funding acquisition, Writing – review and editing; Jonathan S Tsay, Conceptualization, Data curation, Formal analysis, Supervision, Funding acquisition, Investigation, Writing – original draft, Writing – review and editing

### Author ORCIDs

Tianhe Wang ⓘ https://orcid.org/0000-0002-0131-850X
Richard B Ivry ⓘ https://orcid.org/0000-0003-4728-5130
Jonathan S Tsay ⓘ https://orcid.org/0000-0002-3992-9023

### Ethics

Human subjects: All experimental protocols were approved by the Institutional Review Board at the University of California, Berkeley (Approval number: 2016-02-8439). Informed consent was obtained from all participants.

Reviewer #1 (Public review): https://doi.org/10.7554/eLife.100715.4.sa1
Reviewer #2 (Public review): https://doi.org/10.7554/eLife.100715.4.sa2
Reviewer #3 (Public review): https://doi.org/10.7554/eLife.100715.4.sa3
Author response https://doi.org/10.7554/eLife.100715.4.sa4

## Additional files

### Supplementary files
Supplementary file 1. Parameter estimates from best fits using the group-level data for the TR +TG model from Experiments 1b and 3b. See 'Methods' for description of each parameter. (a) Participant moved on the trackpad in Exp 3b. We assumed the movement distance was 1 cm and scaled the parameters accordingly. (b) The estimate of $y_r$ is much smaller in Exp 3b compared to Exp 1b, suggesting the workspace in Exp 3b is closer to the body. This attenuates the average magnitude of the bias.

MDAR checklist

### Data availability
Data and code is available at https://github.com/shion707/Motor-Bias, copy archived at *Wang, 2026*.

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

## Appendix 1

We present here a formalized mathematical analysis to illustrate how different models produce different numbers of peaks in the movement-bias function. Across all models, the motor bias $B(\theta)$ represents how much the hand deviates perpendicularly from the intended reach direction $(\hat{t}_\perp)$, relative to the target distance $(R)$:

$$B(\theta) \approx \delta\theta(\theta) = \frac{\mathbf{b}(\theta) \cdot \hat{\mathbf{t}}_\perp}{R}$$

where $\mathbf{b}(\theta)$ denotes a small bias vector field describing systematic deviations in endpoint position as a function of movement direction $\theta$.

The number of peaks observed in the motor bias function, $E(\theta)$, is determined by the angular periodicity of $\mathbf{b}(\theta)$. We outline below how the periodicity of $\mathbf{b}(\theta)$ can be estimated for each model.

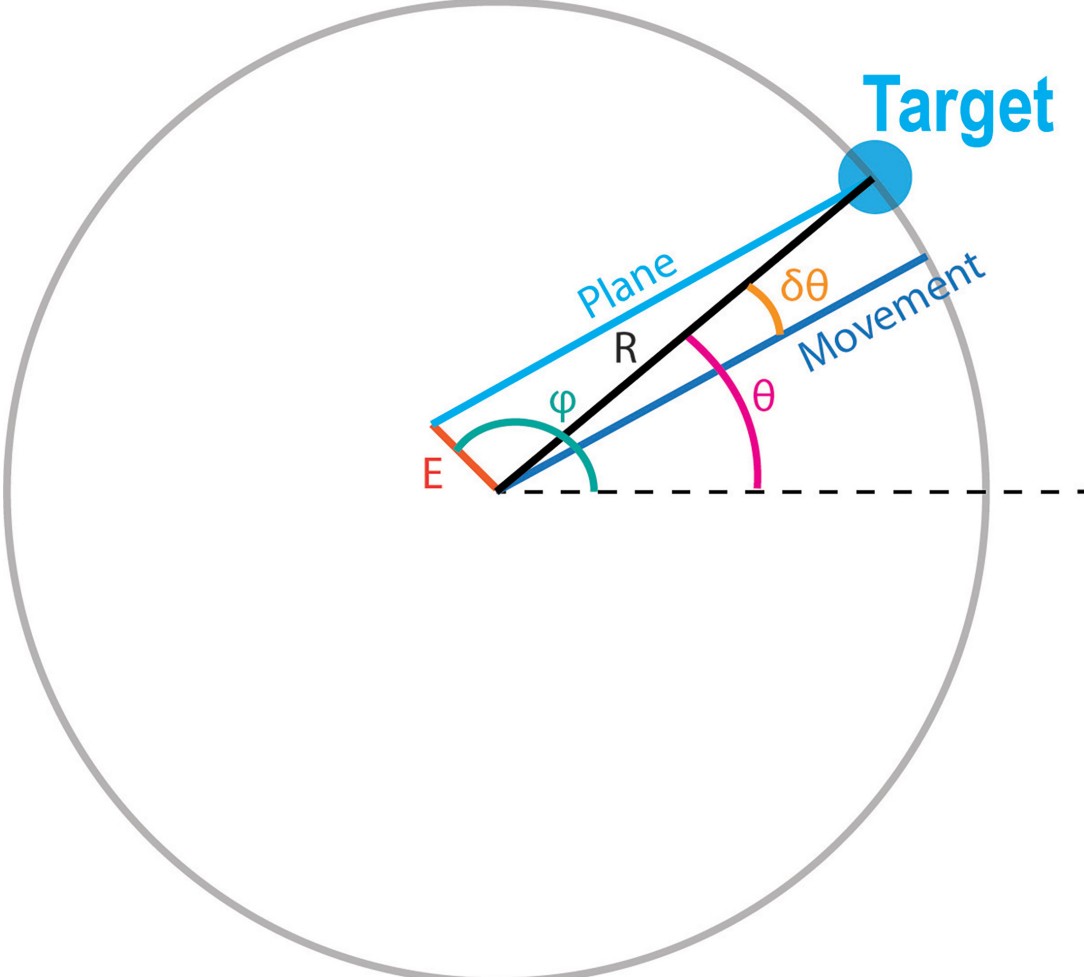

**Appendix 1—figure 1.** Annotation of the Proprioceptive Bias model.

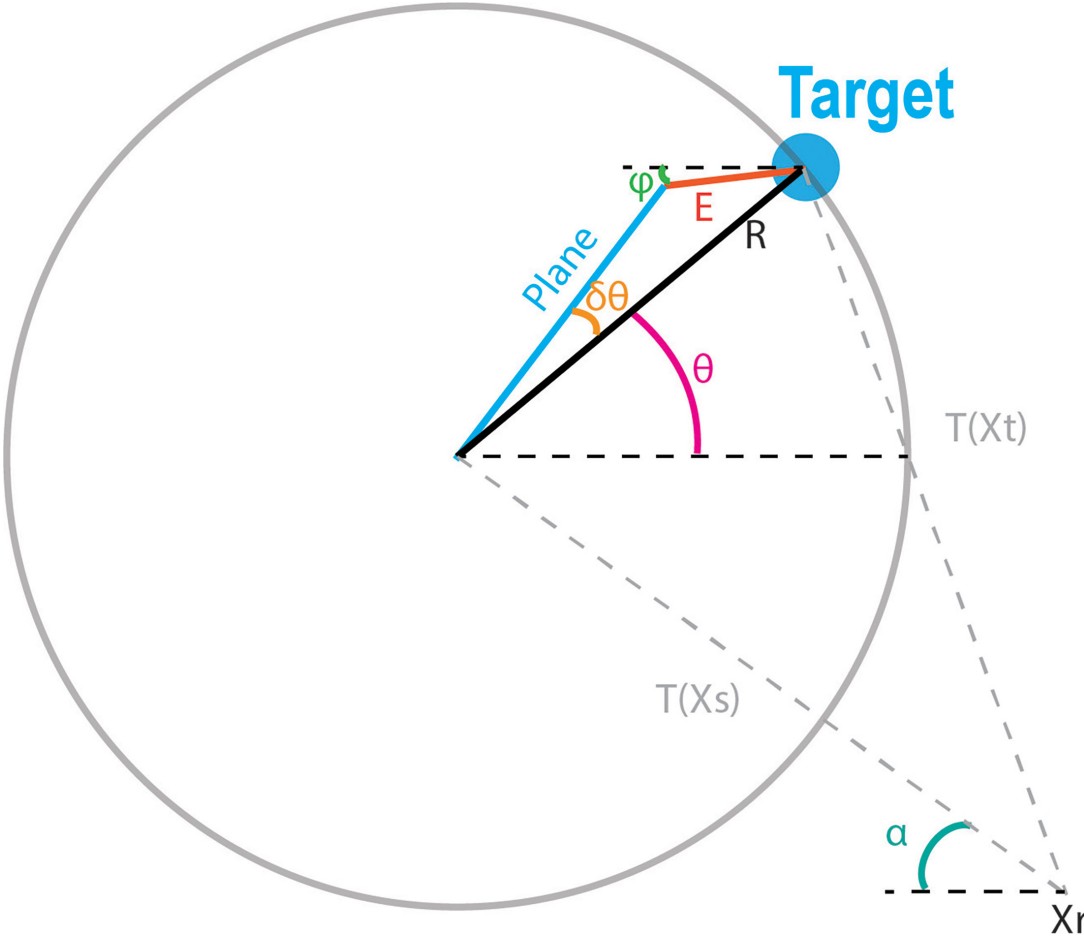

**Appendix 1—figure 2.** Annotation of the Transformation Bias model.

## Target bias model

The perceived target locations are attracted toward the diagonal axes (45°, 135°, 225°, 315°). As the target direction $\theta$ spans 0°–360°, this diagonal attraction produces a motor-bias pattern with a periodicity of 90°, resulting in a four-lobe waveform (i.e., four peaks and four troughs).

## Proprioceptive bias model

The perceived starting point of the hand is biased by a constant vector $\boldsymbol{e} = (x_e, y_e)$. The planned movement vector then becomes the difference between the target and the biased start position $(\boldsymbol{t} - \boldsymbol{e})$. Given the variable definitions in the **Appendix 1—figure 1** below, the resulting angular motor bias can be expressed as

$$sin\left(\delta\theta\left(\theta\right)\right) = \frac{c * sin\left(\phi_e - \theta\right)}{\sqrt{R^2 + e^2 - 2R * e * cos\left(\phi_e - \theta\right)}}$$

where $\phi_e$ denotes the direction of the proprioceptive bias. Assuming that the proprioceptive bias magnitude $e$ at the start position is much smaller than the target distance $R$, this simplifies to

$$\delta\theta\left(\theta\right) \approx arccos\sqrt{\frac{1 - cos\left(\phi_e - \theta\right)}{2}}$$

This function has one peak within 0°–360°.

## Transformation bias model

The transformation bias is modeled as a radial vector field

$$\mathbf{T(x)} = \alpha\left(\|\mathbf{x} - \mathbf{x}_r\|\right) \hat{\mathbf{r}}$$

centered at a reference point $x_r$ near the shoulder. When both the start position $(\mathbf{x}_s)$ and the target position $(\mathbf{x}_t)$ are encoded visually, the angular motor bias is determined by the difference between the two transformed vectors:

$$\mathbf{E} = \alpha\left(\|\mathbf{x}_t - \mathbf{x}_r\| - \|\mathbf{x}_s - \mathbf{x}_r\|\right) \hat{\mathbf{r}}.$$

From the geometry in the *Appendix 1—figure 2*, $\| \mathbf{x}_t - \mathbf{x}_r \| - \| \mathbf{x}_s - \mathbf{x}_r \|$ can be approximated as $c * sin\left(\theta - \alpha\right)$, where $\alpha$ is set by the relative positions of the start and reference points, and $c$ is a constant. The motor bias can then be written as

$$sin\left(\delta\theta\left(\theta\right)\right) \approx \frac{esin\left(\theta - \phi\right)}{R} = \frac{csin\left(\theta - \alpha\right)sin\left(\theta - \phi\right)}{R} = \frac{c\left(cos\left(\alpha - \phi\right) - cos\left(2\theta - \phi - \alpha\right)\right)}{2R}.$$

Given that $\delta\theta\left(\theta\right)$ is small, $\delta\theta\left(\theta\right) \approx sin\left(\delta\theta\left(\theta\right)\right)$, so it has two peaks and two valleys within 0°–360°.

