## [Editor Report · eLife Assessment]

This **important** study uses an original method to address the longstanding question of why reaching movements are often biased. The combination of a wide range of experimental conditions and computational modeling is a strength. **Convincing** evidence is presented in support of the main claim that most of the biases in 2-D movement planning originate in misalignment between visuo-proprioceptive reference frames.

---

## [Referee Report · Reviewer #1 (Public review)]

Wang et al. studied an old, still unresolved problem: Why are reaching movements often biased? Using data from a set of new experiments and from earlier studies, they identified how the bias in reach direction varies with movement direction and movement extent, and how this depends on factors such as the hand used, the presence of visual feedback, the size and location of the workspace, the visibility of the start position and implicit sensorimotor adaptation. They then examined whether a target bias, a proprioceptive bias, a bias in the transformation from visual to proprioceptive coordinates and/or biomechanical factors could explain the observed patterns of biases. The authors conclude that biases are best explained by a combination of transformation and target biases.

A strength of this study is that it used a wide range of experimental conditions with also a high resolution of movement directions and large numbers of participants, which produced a much more complete picture of the factors determining movement biases than previous studies did. The study used an original, powerful and elegant method to distinguish between the various possible origins of motor bias, based on the number of peaks in the motor bias plotted as a function of movement direction. The biomechanical explanation of motor biases could not be tested in this way, but this explanation was excluded in a different way using data on implicit sensorimotor adaptation. This was also an elegant method as it allowed the authors to test biomechanical explanations without the need to commit to a certain biomechanical cost function.

Overall, the authors have done a good job mapping out reaching biases in a wide range of conditions, revealing new patterns in one of the most basic tasks, and the evidence for the proposed origins is convincing. The study will likely have substantial impact on the field, as the approach taken is easily applicable to other experimental conditions. As such, the study can spark future research on the origin of reaching biases.

Comments on revisions:

The authors have addressed my concerns convincingly. The inclusion of the data on movement extent, and the comparison with the data and explanation of Gordon et al. (1994), has strengthened the paper, as it shows that the proposed model can also explain biases in movement extent. I also appreciate the addition of the mathematical analysis, although I suspect that this analysis can be developed further to yield more detailed insights into the conditions under which the 1-, 2- and 4-peaked patterns arise, but that is a more suitable question for follow-up work.

---

## [Referee Report · Reviewer #2 (Public review)]

Summary:

This work examines an important question in the planning and control of reaching movements - where do biases in our reaching movements arise and what might this tell us about the planning process. They compare several different computational models to explain the results from a range of experiments including those within the literature. Overall, they highlight that motor biases are primarily caused errors in the transformation between eye and hand reference frames. One strength of the paper is the large numbers of participants studied across many experiments. However, one weakness is that most of the experiments follow a very similar planar reaching design - with slicing movements through targets rather than stopping within a target. This is partially addressed with Exp 4. This work provides a valuable insight into the biases that govern reaching movements. While the evidence is solid for planar reaching movements, further support in the manner of 3D reaching movements would help strengthen the findings.

Strengths:

The work uses a large number of participants both with studies in the laboratory which can be controlled well and a huge number of participants via online studies. In addition, they use a large number of reaching directions allowing careful comparison across models. Together these allow a clear comparison between models which is much stronger than would usually be performed.

Comments on revisions:

I thank the authors for all the additions to the manuscript, which has addressed my concerns.

---

## [Referee Report · Reviewer #3 (Public review)]

This study makes excellent use of a uniquely large dataset of reaching movements collected over several decades to evaluate the origins of systematic motor biases. The analyses convincingly demonstrate that these biases are not explained by errors in sensed hand position or by biomechanical constraints, but instead arise from a misalignment between eye-centric and body-centric representations of position. By testing multiple computational models across diverse contexts-including different effectors, visible versus occluded start positions-the authors provide strong evidence for their transformation model. My earlier concerns have been addressed, and I find the work to be a significant and timely contribution that will be of broad interest to researchers studying visuomotor control, perception, and sensorimotor integration.

Comments on revisions:

None

---

## [Author Response]

The following is the authors’ response to the previous reviews

**General recommendations (from the Reviewing Editor):**
The reviewers agreed that addressing some specific concerns would improve the clarity of the paper and the strength of the conclusions. These points are listed below, and described in more detail in the reviewer-specific 'Recommendations for Authors':

We thanks the editor and reviewers for the encouraging feedback and constructive comments. We provide our point-by-point response below.

(1) The details of the new experiment including number of subjects and a description of the analysis should be provided in the main text.

We now provide a detailed description of the methods (including the number of subjects; N = 30) and analyses for the new experiment. See our response to Reviewer 2 for more details.

(2) It would be informative to see how the amplitude biases observed, agree with those found by Gordon et al. 1994.

Addressed. Please see our response to Reviewer 1, comment 1.

(3) Each of the models lead to different bias patterns. It would be very helpful to hear the author's interpretation, ideally with a mathematical explanation, of what leads to these distinct patterns.

Addressed. Please see our response to Reviewer 1, comment 2.

**Reviewer #1 (Recommendations for the authors):**
(1) Most of my points have been addressed convincingly in this revision. The new experiment in which also biases in movement amplitude were determined is a welcome addition to the paper. However, I could not see the results of this study, as the authors did not include Fig. 4 in the manuscript, but repeated Fig. 3. That's unfortunate as I would have like to see the similarity between the biases in direction and amplitude. Moreover, I would have liked to see how the amplitude biases agree with those found by Gordon et al. EBR (1994) 99:112-130, and to which extent Gordon et al.'s explanation can explain the pattern.

We apologize for including the incorrect figure in the previous version of our manuscript. We did make a correction and submitted a corrected version, but it appears that it didn’t make its way to you. The correct Figure 4 is now in the manuscript.

The motor biases in amplitude (extent) observed in Experiment 4 (Author response image 1) are qualitatively similar to the pattern reported by Gordon et al. 1994. While the exact peaks do not match perfectly, both datasets show a two-peaked pattern.

Gordon et al. (1994) attributed the bias in amplitude to direction-dependent variation in movement speed which, in their view, arise from anisotropies in limb inertia. Specifically, moving the upper arm along its quasiorthogonal direction (i.e., rotation about the elbow) requires lower effective inertia than moving parallel to the upper-arm axis. Given the arm posture in both datasets, the upper limb points toward ~135°/315°, with the orthogonal direction corresponding to ~45°/225°. The two-peaked speed profiles in both our data Author response image 1 and Gordon et al. are consistent with this prediction.

**Author response image 1. sa4fig1:** 

Gordon et al (1994) noted that, while the extent bias function should mirror the speed bias function, the motor planning system might proactively compensate for the speed bias. Indeed, while the extent and speed bias functions are roughly aligned in their study, the two are misaligned in our Experiment 4. For example, the speed function peaks around 45° which corresponds to a valley in the extent bias function. The difference between their data and ours could be due to a difference in the starting point configuration. However, their model predicts alignment of the speed and extent functions independent of starting point configuration. In contrast, the TR+TG model does predict our observed extent bias function and yields predictions about how this should change with different start point configurations. As such, while heterogeneity in movement speed may contribute to extent bias to some degree, we think the transformation bias and visual-target bias likely play a larger role in determining the amplitude bias observed extent bias at movement endpoint.

We have added a discussion section about the bias function reported by Gordon et al. (1994) and their account in the manuscript (lines 482-493). We do not repeat it here, as the content largely overlaps with the response above.

(2) One of the most important new insights from this study is that the three single-source models lead to different bias patterns, with 1, 2 or 4 peaks. However, what I miss in the paper is an intuitive explanation why they do so. Now, the models are described and their predictions are shown, but it remains unclear where these distinct patterns come from. As scientists, we want to understand things, so I would very much appreciate if the authors can provide such an intuitive explanation, for instance using a mathematical proof. That could also identify how general these patterns are, or if there are certain requirements for them to occur (such as a certain shape of the transformation bias).

Note that the closed-form mathematical expression for the motor bias function is not straight forward. As such, the intuition comes primarily from inspection, that is, the model simulations themselves, what we show Figure 1 of the paper. Importantly, the model predictions are insensitive to the parameter values over a reasonable range. Thus, the number of peaks predicted by each model is a core distinguishing feature. We present in the Supplementary Results a formalized mathematical analysis to illustrate how different models produce different numbers of peaks in the movement-bias function.

(3) I think it's a good idea to change the previous "Visual Bias" into a "Target Bias". This raises the question whether the "Prioprioceptive Bias" should not be changed into a "Hand Bias" or "Start Bias"?

While we appreciate the reviewer’s point here, we prefer the term “Proprioceptive Bias” given that this term has been used in the literature and provides a contrast with sources of bias arising from vision. “Hand Bias” and "Start Bias” seem more ambiguous.

L51: I think "would fall short" should be replaced by "would overshoot".L127: I think "biased toward the vertical axis" should be replaced by "biased away from the vertical axis". Figure 3 still contains the old terminology like T+V. Please replace by the new terminology. L255: Replace "Exp 1a" by "Exp 1b".L376: Replace 60 by 6.L831-2: I hope the summed LL was maximized, not minimized.

Thanks for catching the typos. We have corrected all of them.

**Reviewer #2 (Recommendations for the authors):**
I think that Experiment 4 does not mention how many participants performed the study. (Only in the response to the reviewers I found this)

We have added information regarding the number of participants in the Fig 4 (N=30).

I am very happy that the authors added the biomechanical simulation into the paper. I am not convinced that this addressed my concerns exactly but it is an excellent addition and the authors have now adjusted the text appropriately.

We appreciate the positive response to our additional assessment of biomechanical factors. We welcome any additional information on how we might fully address this issue.

line 826: extend -> extent

Corrected.

Figure 4. I think that the authors have put the wrong figure here. I cannot see any data for extent. I would need to see this figure or please correct me - but the caption doesn't match the figure and I don't see the results clearly. (I think the review might have the correct figure).

We apologize for this mistake. We now provided the correct Figure 4 in the paper (also included in the first page of the response letter).

I am missing the detailed description on when the direction error and distance error were calculated for exp 4 - and what exactly was used? How did the authors examine the values without correction? What time point was used? Did I miss the analysis section for this?

Participants were instructed to make fast, straight movement without any corrections and were given up to 1 s to complete the movement. Hand position was recorded once the movement speed dropped below 1 cm/s. On 99.8% of trials, movement speed did not increase once this threshold was passed, indicating that the participants adhered to the instructions. On the remaining trials, we detected a secondary corrective movement (increase in speed >5 cm/s). On these trials, we used the position recorded when the movement speed initially dropped below 1 cm/s as the endpoint position. The pattern of results would be the same were we to exclude these trials.

This information has been added to the Methods section (line 661-666).